# Towards Coordinate- and Dimension-Agnostic Machine Learning for Partial Differential Equations

## Abstract

Machine learning methods for data-driven identification of partial differential equations (PDEs) are typically defined for a fixed number of spatial dimensions and a particular choice of coordinates in which the data have been collected. This dependence prevents the learned equation from generalizing to other spaces. In this work, we reformulate the problem in terms of *coordinate-* and *dimension-independent* representations, paving the way toward what we might call "spatially liberated" PDE learning. In this work, we propose an approach to learning PDEs by expressing them in a way that is independent of the coordinate system and even the underlying manifold where the equation is defined. This allows us to learn a PDE in low-dimensional spaces and generalize to higher-dimensional spaces with different geometric properties. We provide extensive numerical experiments that demonstrate that our approach allows for robust transferability across various geometric contexts. We show that the dynamics learned in one space can be used, without retraining, to make accurate predictions in other spaces with different dimensions, coordinate systems, boundary conditions, and curvatures, by recomputing invariant features.

## 1 Introduction

Machine learning has traditionally been used to identify and model partial differential equations (PDEs) from observational data (Krischer et al., 1993; González-García et al., 1998; Blechschmidt & Ernst, 2021; Cuomo et al., 2022). In particular, deep learning techniques such as physics-informed machine learning (Karniadakis et al., 2021; Raissi et al., 2019) and operator learning (Lu et al., 2021; Kovachki et al., 2023; Fabiani et al., 2025), often in combination with symbolic regression techniques (Brunton et al., 2016; Cranmer, 2023), can extract spatiotemporal patterns from experimental measurements and (approximately) infer the underlying physical laws. This has been especially useful for describing complex dynamical systems in various scientific and engineering domains, where conventional numerical methods can be computationally expensive (Kochkov et al., 2021), or where we only have access to an incomplete description of the system (Martin-Linares et al., 2023). As a result, learning-based PDE models are used on a wide range of scales, from fine-grained descriptions at the particle level (Lee et al., 2023) to coarse-grained representations in terms of density fields (Psarellis et al., 2024), providing a data-driven alternative to traditional physics-based derivations.

Real-world data are often expressed in domains and coordinate systems that are determined by the experimental acquisition process. For instance, bacterial chemotaxis can be captured as waves in effectively one-dimensional channels (Adler, 1966), or visualized as two-dimensional images — top-down, with upright (Phan et al., 2020) or bottom-up, with inverted microscopy (Morris et al., 2017). All three experimental settings produce data of the same physical process supported in different geometries and coordinate systems. The conventional machine learning approaches for PDE identification are inherently tied to a predefined spatial framework, preventing them from generalizing the learned physical mechanism to different coordinate systems, geometries, or dimensions. The recent work of Psarellis et al. (2024) has proposed a coordinate-independent PDE learning approach using exterior calculus. Their method allows them to learn a time evolution scalar PDE independently of the coordinate system, and they show that the learning generalizes to different coordinate system representations of the same domain.

In this work we push these ideas substantially further, and show that we can learn PDEs not just in a coordinate-independent way but also independently of the data domain, dimension, and geometry.

> *We propose a framework that allows us to learn a data-driven representation of a PDE operator in one domain, and then make accurate predictions in another, when the underlying physics is the same.*

We are able to do this by combining ideas from the coordinate-free learning of time evolution scalar PDEs from Psarellis et al. (2024), and by recent work in *any-dimensional learning* that proposes a theoretical framework for machine learning models that are defined on a fixed set of parameters but can be trained and evaluated in spaces of arbitrary size (Levin & Díaz, 2024; Levin et al., 2025).

The most widely used any-dimensional models are arguably graph neural networks (GNNs), where a model defined on a fixed set of parameters can be trained and evaluated on graphs of any size (and GNNs are completely agnostic to the graph sizes). In order to have a sufficiently expressive machine learning model with a fixed number of parameters while the dimension of the input space grows arbitrarily, one typically relies on some form of symmetry constraint (this is mathematically formalized via representation stability (Farb, 2014)). For example, graph neural networks are equivariant with respect to node permutations.

In this work we learn PDE operators that are invariant with respect to local orthogonal symmetries (Weyl, 2015; Frankel, 2011; Nakahara, 2018; Abraham et al., 2012; Villar et al., 2023), which is a feature observed in many natural phenomena. For these systems, we show how machine learning can achieve coordinate- and dimension-agnostic transferability, enabling training in any space and coordinates, yet being able to make accurate predictions across all others, regardless of coordinates, but also dimensions, boundary conditions, and spatial *intrinsic* curvatures (i.e. Riemann curvature tensors (Do Carmo & Flaherty Francis, 1992)). Fusion of training data obtained at different coordinates/geometries is also enabled.

In our experiments, we use neural networks to identify time evolution PDEs of coupled scalar fields (Section 3.1), general scalar PDEs (Section 3.2), and vector PDEs (Section 3.3) in a coordinate-free and dimension-free way. In particular, we apply our models to the FitzHugh-Nagumo (Bär et al., 2003) and Barkley (Barkley, 1991) reaction-diffusion model operators; and a modified Patlak-Keller-Segel model, derived from *in-situ* observations of chemotactic bacteria (Phan et al., 2020). We also illustrate our method on the Helmholtz equation, and the incompressible Navier-Stokes equation. Most of our models are trained on data generated in a periodic one-dimensional space using Cartesian coordinates, and we then demonstrate their ability to generalize to higher-dimensional spaces. These include two- and three-dimensional Euclidean spaces in Cartesian or non-Cartesian coordinates, as well as a two-dimensional submanifold of a sphere in Euclidean space, described using geographic as well as stereographic coordinates. We will also consider different boundary conditions in these new spaces, such as a no-flux Neumann boundary; we will provide a quantitative analysis of the generalization and robustness properties of our approach in Appendices K and L.

## 2 COORDINATE-FREE OPERATOR LEARNING ACROSS DIMENSIONS

Many differential equations can be formulated in a coordinate-free and domain-free mathematical expression. For example, the heat equation written as $\frac{\partial u}{\partial t} = \sum_{i=1}^{n} \frac{\delta^2 u}{\delta x_i^2}$ is not coordinate free, since it is expressed in terms of local coordinates. However the heat equation expressed as $\frac{\partial u}{\partial t} = \Delta u$ (where $\Delta$ is the Laplacian operator of the Riemannian manifold $\mathcal{M}$ where $u$ is defined) is not only coordinate-free, since the Laplacian $\Delta$ is an intrinsic operator of $\mathcal{M}$, but also it is domain-free, since the Laplacian exists for every Riemannian manifold. Therefore, the equation is well defined and can be evaluated for manifolds of any dimension. This procedure can be formalized in terms of differential forms and exterior differential systems (see Appendix A). In this paper we argue that we can learn the PDE in one domain and transfer its learning to other manifolds of different dimensions, as long as the PDE is defined as a "dimension-agnostic" expression.

PDEs coming from mathematical physics and other fields of science and engineering often can be expressed in a coordinate-free manner in terms of their independent variables and their (unknown) fields. Notable examples are the Navier-Stokes (Wilson, 2011) and, more generally, the equations

of continuum mechanics (Marsden & Hughes, 1994), the Fokker-Planck equation (Masoliver et al., 1987), the Kuramoto-Sivashinsky equation (Frankel & Sivashinsky, 1987), and many others.

In this paper we consider systems of scalar and vector PDEs. All of them governed by equations that are invariant under local orthogonal transformations (rotations and reflections) and under diffeomorphic translations (Frankel, 2011; Nakahara, 2018; Abraham et al., 2012). Such systems arise frequently in nature, as microscopic physical interactions typically exhibit these symmetries; thus, the emergent PDEs inherit them at macroscopic scales. We consider the PDE to involve at most second-order spatial-derivatives, in which we adopt the power-counting scheme of effective field theory (Kaplan, 2005; Buchalla et al., 2014) with the total differential order of each term defined as the sum of its derivative orders.[1] Higher order features are feasible via a recursive construction; see O for the construction, and a 4-th order example. We then observe that when we apply these restrictions, the resulting class of PDEs turn out to be domain-free and dimension-free and we can therefore generalize the learning to other domains in other dimensions.

Our approach can be summarized in two main steps (see Figure 1):

1. Given data satisfying a PDE, first learn a spatially coordinate-free representation of the PDE.
2. Given a new set of initial and/or boundary conditions in a possibly different coordinate system, or even dimension, use a coordinate-free solver of the PDE to evolve the dynamics.

Coordinate-free integrators are a classical subject of study in numerical analysis (Dinesh et al., 2000).

**Generators of invariant fields.** Given $N$ scalar fields $\{\Psi_1, \ldots, \Psi_N\}$ (i.e. 0-forms on a Riemannian manifold $\mathcal{M}$) the set of all possible orthogonally symmetric operators from 0-forms to 0-forms involving up to second-order derivatives can be obtained from[2]:

- *The fields* $\Psi_j$ where $j = 1, \ldots, N$.
- *The Laplacians* $\Delta\Psi_j$, where $j = 1, \ldots, N$.
- *The inner products of the gradients* $\langle \boldsymbol{\nabla}\Psi_j, \boldsymbol{\nabla}\Psi_k \rangle$, where $j, k = 1, \ldots, N$. Since these are symmetric, there are $N(N+1)/2$ distinct products.

The Laplacians and the inner products of the gradients above are defined with respect to the Riemannian metric of $\mathcal{M}$, and can be expressed in coordinate-free form using the metric structure of the manifold (see Appendix A).

This yields a basis of invariant features $\mathcal{B}(\Psi_1, \ldots, \Psi_N)$ (or $\mathcal{B}$ when the input is clear by context),

$$\mathcal{B}(\Psi_1, \ldots, \Psi_N) = ([\Psi_j]_{j=1}^N, [\Delta\Psi_j]_{j=1}^N, [\langle\nabla\Psi_j, \nabla\Psi_k\rangle]_{1\le j\le k\le N}) =: \left(b_1, \ldots, b_{\frac{N(N+5)}{2}}\right). \quad (1)$$

The invariant features (fields, gradient inner products, and Laplacians) are computed pointwise from data. This can be achieved in practice via finite differences, or alternatively through automatic differentiation or Fourier features, depending on the type of integrator used; see Equations 40–43 and 59–63 in the appendix for explicit formulas.

**Coordinate-free learning of evolution PDEs.**

Given $\Psi_1, \ldots, \Psi_N$ as above we consider the PDE identification problem

$$\frac{\partial\Psi_j}{\partial t} = \mathcal{F}_j(\Psi_1, \ldots, \Psi_N, t), \quad j = 1, \ldots, N \quad (2)$$

---

[1] This is the power-counting scheme for wave numbers, which is of critical importance when creating an effective field theory for coarse-grained dynamics. Since each spatial derivative introduces a factor of the wave number (e.g. in Fourier space), higher-order terms are increasingly suppressed at low wave numbers. By explicitly tracking the total derivative order, this method provides a more accurate characterization of the dominant large-scale emergent behavior of the system.

[2] This follows from the first fundamental theorem of classical invariant theory for the orthogonal group (Weyl, 1946; Spivak, 1999), which has been extensively used in machine learning (Villar et al., 2021; Gregory et al., 2024). All SO($n$)-scalars can be built locally from derivatives of scalars, the metric (Kronecker delta), and the Levi-Civita symbol via basic arithmetic operations and contractions (Spivak, 1999); for O($n$)-invariance the Levi-Civita symbol is excluded. Our claim comes from applying this theorem to the second-order jet space, in which in this work we restrict to scalar expressions with total order of derivatives at most two. For higher-order, we can list all possible features constructed from the metric, the fields, and its derivatives — up to the desired order — to be used as invariant features.

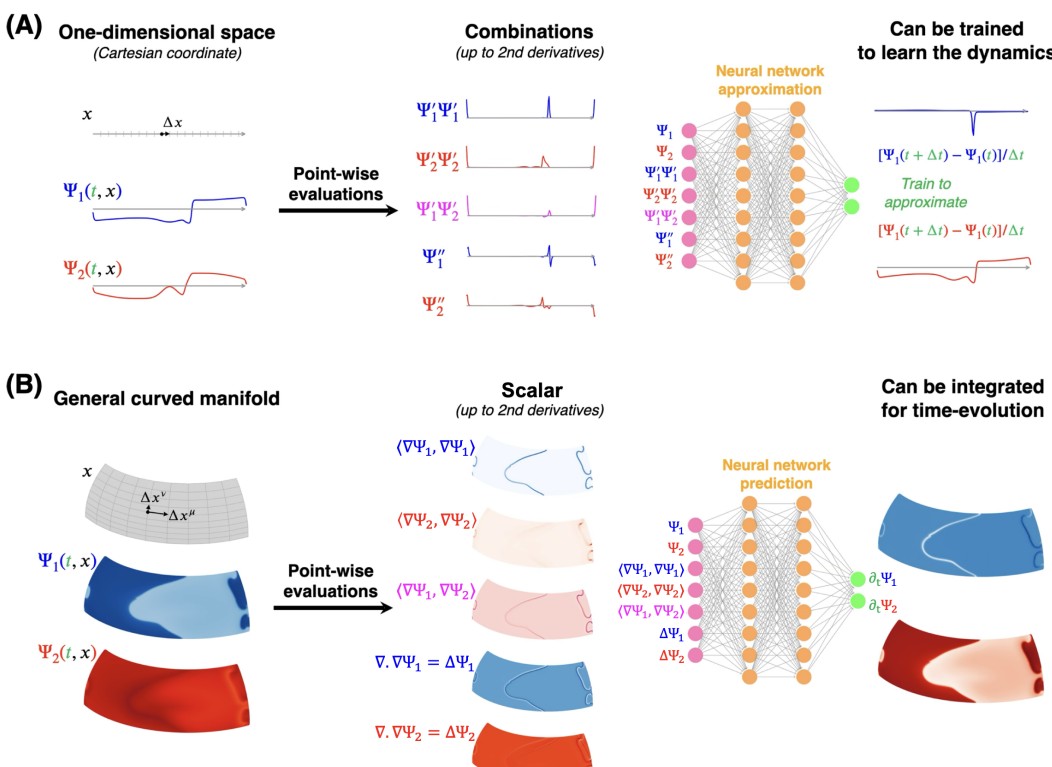

**Figure 1:** We aim to demonstrate that it is possible to approximate the dynamics of a class of evolutionary PDE systems in a coordinate- and dimension-free manner using neural networks. We package the input in a coordinate- and dimension-free manner. **(A)** We learn the one-dimensional space right-hand-side operator of the PDEs in a Cartesian coordinate. **(B)** We can then simulate the dynamics in arbitrary coordinates and dimensions (the figure illustrates the case of two-dimensional curved space).

---

**Algorithm 1** Any-dimensional PDE identification

1: **Input:** Discrete samples of scalar fields $\Psi_j(x,t)$, $j = 1, \ldots, N$ where $x \in \mathcal{M}$ a Riemannian manifold and $t \in \mathbb{R}$.
2: **Output:** Neural networks $\hat{\mathcal{F}}_j$ approximating the solution of $\frac{\partial \Psi_j}{\partial t} = \hat{\mathcal{F}}_j(\mathcal{B}(\Psi_1, \ldots, \Psi_N))$ where $\mathcal{B}(\Psi_1, \ldots \Psi_N)$ is as a coordinate system for functions defined on $\mathcal{M}$.
3:
4: **for** $j = 1, \ldots, N$ **do**
5:     Use finite differences to approximate gradients and Laplacians of the fields $\nabla \Psi_j, \Delta \Psi_j$.
6: **end for**
7: Compute the basis elements $\mathcal{B}(\Psi_1, \ldots, \Psi_N) = ([\Psi_j]_{j=1}^N, [\Delta \Psi_j]_{j=1}^N, [\langle \nabla \Psi_j, \nabla \Psi_k \rangle]_{1 \le j \le k \le N}) =:$
    $\left( b_1, \ldots, b_{\frac{N(N+5)}{2}} \right)$.
8: Train neural networks to identify the PDE satisfied by the scalar fields, where the input to the neural network is given in the basis $\mathcal{B}$. Namely, $\frac{\partial \Psi_j}{\partial t} = \hat{\mathcal{F}}_j(\mathcal{B}(\Psi_1, \ldots, \Psi_N))$.
9: **Return** $\hat{\mathcal{F}}_j$, $j = 1, \ldots, N$.
10: **Generalization to other domains:** Given scalar fields $\tilde{\Psi}_1, \ldots \tilde{\Psi}_N$ in another Riemannian manifold $\tilde{M}$. We can construct $\mathcal{B}(\tilde{\Psi}_1, \ldots, \tilde{\Psi}_N))$. We say that the $\tilde{\Psi}_j$'s satisfy the same PDE as the $\Psi_j$'s if $\frac{\partial \tilde{\Psi}_j}{\partial t} = \hat{\mathcal{F}}_j(\mathcal{B}(\tilde{\Psi}_1, \ldots, \tilde{\Psi}_N))$. The learning of $\tilde{\mathcal{F}}_j$ in any domain generalizes to *the same PDE* in any other domain.

where we assume the $\mathcal{F}_j$'s are (possibly non-linear) orthogonally symmetric second order operators defining the right-hand-side of the evolutionary PDE. We consider neural networks that learn $\hat{\mathcal{F}}_j(\mathcal{B})$ as functions of the features (1), such that the integration of the corresponding PDE via a coordinate-free integrator approximates the training data (see Algorithm 1). A **coordinate-free integrator** in this context is one that performs the integration of the PDE in terms of the features $b_i$. Note that this approach can be seen as a form of canonicalization (see Kaba et al. (2023); Olver (2003)), where the input data is transformed into a set of canonical invariants (here $\mathcal{B}$) to perform the learning.

If an equation contains variable spatio-temporal coefficients, these can be treated as additional fields that participate in the construction of the coordinate-free basis. See N for an example.

**Coordinate-free learning of general scalar PDEs.** The time evolution PDEs we consider here can be expressed as $\frac{\partial u}{\partial t} = \mathcal{F}(\mathcal{B}(u))$ where $\frac{\partial u}{\partial t}$ only appears on the left-hand-side. A general PDE, however is an *implicit* algebraic constraint between the elements of $B$. We therefore do not *a priori* know which variables can be written on the left- and right-hand sides of the PDE. Instead, we can utilize manifold learning methods, linear (ae.g. PCA) for linear PDEs, nonlinear (e.g. autoencoders or kernel-PCA) for nonlinear PDEs, to discover the *implicit* constraint within the coordinate-free basis. In this way, the PDE operator is represented by a latent model constraining the relationship between the variables in $\mathcal{B}$. After this is learned, it can be 'extracted' symbolically using symbolic regression (see Appendix N) for an example, or used implicitly by numerical solvers. Of course, there is the caveat that, when the PDE is nonlinear, the constraint learned is only guaranteed to be accurate in the region adequately sampled by the training data.

The workflow for a general PDE is as follows:
1. Project the data satisfying the PDE, $u$, to the coordinate- and dimension-free basis $\mathcal{B}(u)$.
2. Use manifold-learning to parametrize the submanifold defined by the data in terms of $\mathcal{B}(u)$.
3. Use the trained model as a representation of the operator on different domains, or extract a symbolic representation of the operator using symbolic regression, before transferring to another domain or dimension.

We implement this method for the Helmholtz equation in Section 3.2.

We remark that, in order to solve a higher-dimensional problem, we must have information about the 'well-posedness' of that problem. For example, the harmonic oscillator needs only two scalar initial/boundary conditions to be solved, whereas the Helmholtz equation needs continua of boundary conditions to be well-posed. The extent of external (e.g. boundary) information necessary for a PDE to be well-posed cannot be encoded in our coordinate-free representation of the operator in general.

**Extension to vector PDEs.** Our proposed approach can be extended to vector PDEs, or more generally *coupled scalar and vector field PDEs*, by computing an extended coordinate-free feature basis $\tilde{\mathcal{B}}$, which includes scalar features $\mathcal{S}$ and vector features $\mathcal{V}$. Given this basis, a general class of coupled scalar-vector PDEs takes the form:

$$\mathcal{F}(\tilde{\mathcal{B}}) = \sum_{i=1}^{|\mathcal{V}|} f_i\left(\{s_j\}_{j=1}^{|\mathcal{S}|}\right) v_i = 0 \tag{3}$$

where $f_i : \mathcal{S} \to \mathbb{R}$ are arbitrary nonlinear functions. This equation matches the parameterization of *equivariant* functions to actions by the orthogonal group from Villar et al. (2021). We bound the degree of the differentials in the basis by 2 in our work; this can, however, be relaxed based on the problem at hand.

The methodology for constructing coordinate-free features from a scalar field $s$ remains the same as in (1). In particular, the scalar features up to order-2 will be $\{s, \Delta s\} \in \mathcal{S}$, and the sole vector feature will be $\nabla s$. For a vector feature $v$ we will have $\langle v, v \rangle \in \mathcal{S}$ and $v, \Delta v \in \mathcal{V}$, where the latter operator is the component-wise Laplacian. Finally, we include terms of the form $\langle v, \nabla \cdot v \rangle \in \mathcal{V}$, which can be formulated in a cordinate-free manner in terms of the Lie derivative, and terms of the form $\nabla \cdot v \in \mathcal{S}$, denoting divergence. The remaining elements of the library are then computed using the inner product, divergence, and Lie-derivative operations, all of which have coordinate-free formulations in terms of exterior algebra.

**Any-dimensional coordinate-free learning.** The Laplacians and inner products are defined as functions of the input scalar fields for any domain –any Riemannian manifold $(\mathcal{M}, g)$ of any dimension $n$. Therefore $\mathcal{B}$ (or analogously $\tilde{B}$ in the vector PDE case) is defined via a domain- and dimension-

independent description. The learned PDEs are defined as functions of $N(N+5)/2$ features, $(b_1, \ldots, b_{N(N+5)/2})$, independently of the size of the domain of the $b_i$'s (and thus independently of the dimension of the domain of the inputs).

## 3 COMPUTATIONAL EXPERIMENTS

### 3.1 TIME-EVOLUTION SYSTEMS

In this section, we apply the proposed methodology to three time-evolution PDEs: the FitzHugh-Nagumo, the Barkley, and a modified Patlak-Keller-Segel Systems defined below. Each can be expressed in a coordinate-free and dimension-free form (Appendix C), involving two coupled scalar fields (thus $N = 2$ in Eq. (2) and Eq. (1)). We present selected computational experiments in the main text, while extensive additional studies are included in the appendices. See Appendices E–L for the experimental setup details and quantitative evaluation of the results.

**The FitzHugh-Nagumo System.** In the FitzHugh-Nagumo system (Bär et al., 2003) two scalar fields participate: the excitation variable $V(x, t)$ and the recovery variable $W(x, t)$. The spatiotemporal dynamics of these fields are governed by the set of two coupled PDEs:

$$
\begin{aligned}
\partial_t V(x,t) &= \vec{\nabla}^2 V(x,t) + V(x,t) - \frac{1}{3} V^3(x,t) - W(x,t) \, , \\
\partial_t W(x,t) &= \varepsilon \left[ V(x,t) + \beta - \gamma W(x,t) \right] \, ,
\end{aligned}
\tag{4}
$$

where we set $(\varepsilon, \beta, \gamma) = (0.02, 0, 0.5)$. For this choice of parameters, the system generates spiral waves in two-dimensional space and scroll waves in three-dimensional space, whereas such phenomena do not emerge in dimension one.

**The Barkley System.** The Barkley reaction-diffusion system (Barkley, 1991) models excitable media and oscillatory media. It is often used as a qualitative model in pattern forming systems like the Belousov-Zhabotinsky reaction and other systems that are well described by the interaction of an activator and an inhibitor component. In the simplest case, the Barkley system is described by a set of two coupled PDEs:

$$
\begin{aligned}
\partial_t U(x,t) &= \vec{\nabla}^2 U(x,t) + \frac{1}{\varepsilon} U(x,t) \left[ 1 - U(x,t) \right] \left[ U(x,t) - \frac{V(x,t) + b}{a} \right] \, , \\
\partial_t V(x,t) &= U(x,t) - V(x,t) \, ,
\end{aligned}
\tag{5}
$$

where we choose the parameters $(\varepsilon, a, b) = (0.02, 0.75, 0.02)$ so that this system can generate spiral and scroll waves in two- and three-dimensional spaces respectively.

**A Modified Patlak-Keller-Segel System.** Recent technological advancements have enabled us to directly measure *in situ* the time-evolution dynamics of bacterial density and the concentration of environmental cues to which the bacteria respond, with high precision (Phan et al., 2024). We can therefore develop much more accurate macroscopic mathematical models to describe the collective behavior of bacterial swarms. We consider the *modified* Patlak-Keller-Segel model for bacterial chemotaxis as proposed in Phan et al. (2024), supported by direct *in situ* observations of *E. coli* (*Escherichia coli*) density and Asp (Aspartate) chemoattractants. The dynamics are described by a set of two PDEs, which can be can be written in a non-dimensionalized form as follows:

$$
\begin{aligned}
\partial_t A(x,t) &= \vec{\nabla}^2 A(x,t) - \left[ \frac{A(x,t)}{1 + A(x,t)} \right] \left[ \frac{B(x,t)}{1 + B(x,t)} \right] \, , \\
\partial_t B(x,t) &= \alpha B(x,t) + \vec{\nabla}^2 \left[ D_B B(x,t) \right] \\
&\quad - \vec{\nabla} \left( \chi_0 \left[ \frac{B(x,t)}{1 + B(x,t)/B_h} \right] \vec{\nabla} \left\{ \ln \left[ 1 + \frac{A(x,t)}{K_i} \right] \right\} \right) \, ,
\end{aligned}
\tag{6}
$$

where the parameters are estimated from experiments to be $\alpha \approx 1.5 \times 10^{-4}$, $D_B \approx 0.55$, $\chi_0 \approx 4.8$, $B_h \approx 1.3$, and $K_i \approx 0.27$. The field $A(x, t)$ represents the chemical concentration, and the field

$B(x, t)$ represents the bacterial density. The effective chemosensitivity is denoted by $\chi[B(t)]$ and the perceived chemical signal $\Phi[A(t)]$ is defined as:

$$\chi[B(t)] = \chi_0 \left[ \frac{B(t)}{1 + B(t)/B_h} \right] \ , \ \Phi[A(t)] = \ln \left[ 1 + \frac{A(t)}{K_i} \right] \ . \tag{7}$$

**Method.** To study all these systems, we use the following algorithmic workflow:

1. Given training data of the scalar fields $\{\Psi_i(x, t)\}_{i=0}^{N}$ in *one*-dimensional Euclidean space, we project the data onto the spatial coordinate-free basis $\mathcal{B}$ in (1) and estimate the time derivatives $\{\partial \hat{\Psi}_i / \partial t\}$ using a finite-difference scheme,
2. We train a fully-connected neural network $\hat{\mathcal{F}}_{NN}$ to predict $\{\partial \hat{\Psi}_i / \partial t\}$ from the basis $\mathcal{B}$,
3. Given the trained network $\hat{\mathcal{F}}_{NN}$, we integrate the corresponding scalar fields (now defined on different choices of domain) given initial conditions.

**One-Dimensional Euclidean Space.** We first study the generalization of our coordinate-free representation to arbitrary (nonlinear) parametrizations in the spatial domain. In Figure 2, we demonstrate that the learned representation can be used to accurately predict the time evolution for the Patlak-Keller-Segel system under different choices of coordinates. For details, see Appendix D.1.

**Two- and Three- Dimensional Euclidean Space.** We study the generalization of the learned FitzHugh-Nagumo and Barkley systems in two and three dimensions, represented in Figure 3. Again, we observe agreement between the true and generalized (learned from 1-D) dynamics. We emphasize that the learned operator produces spiral waves, which are a property of solutions in higher dimensions that cannot be inferred from the 1-D solution. For details, see Appendix D.2.

**Generalization to different curvatures and topologies.** The proposed coordinate-free representation generalizes to domains with different curvature *and* topology. In particular, we demonstrate that the learned (from 1-D data) operator accurately predicts the time-evolution of the FitzHugh-Nagumo system on a 2-D domain embedded on the surface of a 3-D ball with nontrivial curvature (see Appendix D.3, Figure 6). In this example, we consider both the geographic and stereographic coordinates of the sphere. In addition, we also consider the case where the domain is a flat punctured disk with nontrivial topology. For details, see Appendix D.4.

### 3.2 GENERAL PDE SYSTEMS

**The Helmholtz Equation.** Given data satisfying the Helmholtz equation $\Delta u = -ku$, for $k = 1$, the goal is to learn the PDE from data in dimension 1 (where it reduces to the harmonic oscillator), and generalize to higher dimensions. The coordinate-free features for *one* scalar field are $\mathcal{B} = \{u, \langle \boldsymbol{\nabla} u, \boldsymbol{\nabla} u \rangle, \Delta u\}$. Since we only have *one* PDE, $\mathcal{B}(u)$ should span a two-dimensional manifold that we call $\mathcal{L}$, and since the PDE is linear, PCA should be able to identify it. At this point, the operator is *implicitly encoded* in the weights and components of the fitted PCA model. (Also note that, after computing derivatives, and embedding in $\mathcal{B}$, we discard all information that has to do with which point corresponds to which solution; they all simply lie on a submanifold in that space.)

Now, suppose that we want to transfer this operator to another domain, for example a 2D unit square equipped with the flat metric. We will do this using a Physics-Informed neural network (PINN) (Raissi et al., 2019), solving the PDE implicitly defined by constraints in the manifold $\mathcal{L}$ and boundary conditions prescribed by the user. In the classical setting, for any given domain with an initial condition, the PINN produces a candidate solution $\hat{u}$. The PDE law and boundary conditions are imposed in the loss function as 'soft' constraints. To train the PINN using the implicit coordinate-free representation of the operator, encoded by the prior PCA components, we proceed as follows:

1. Sample a set of points $\{x\}$ in the domain $\Omega$
2. Embed the candidate solution $\hat{u}$ evaluated at $\{x\}$ into the coordinate-free basis $\mathcal{B}(\hat{u})$. (Here the derivatives are computed using automatic differentiation.)
3. Project this data onto the prior learned PCA components. Denote this by $\pi_{\mathcal{L}} \mathcal{B}(\hat{u})$
4. (Left-)invert this projection[3], obtaining the reconstruction $\hat{u}_{\mathcal{B}} = \pi_{\mathcal{L}}^{-1} \pi_{\mathcal{L}} \mathcal{B}(\hat{u})$.

---

[3]Note that this step will be the same for the more general nonlinear case, but this step is non-trivial depending on the method used. An autoencoder for example has this as part of the architecture, while DMaps require additional computational steps. This step does not have to be explicitly differentiable, for NN training purposes!

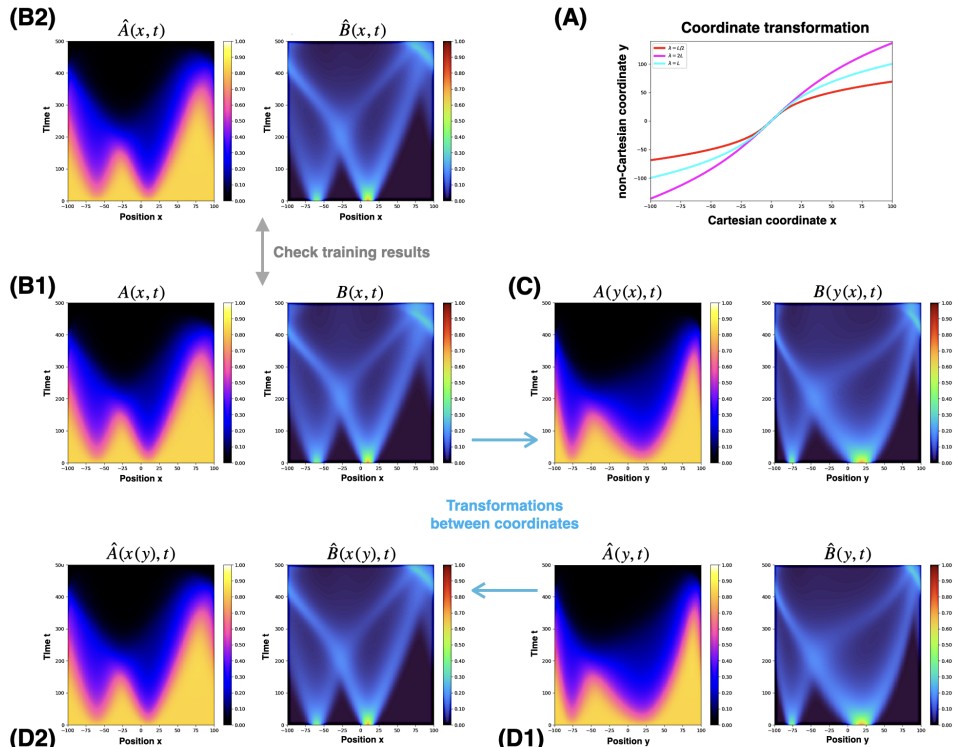

**Figure 2: The Patlak-Keller-Segel field evolution dynamics in a periodic one-dimensional space, with different coordinate systems.** The initial field configurations are the same, up to an extrapolation when transitioning between different coordinate systems. **(A)** The Cartesian and non-Cartesian coordinate systems, using Eq. (25) with different values of $L$ to convert between $x$ and $y$. **(B1)** The time progression of the fields when integrating Eq. (6) to generate the true dynamics Cartesian coordinate $x$. **(B2)** The time progression of the fields when using the trained neural network to generate the learned dynamics in the Cartesian coordinate $x$. **(C)** Transformation of Fig. (B1) into the non-Cartesian coordinate $y$. **(D1)** The time progression of the fields when using the trained neural network to generate the learned dynamics in the non-Cartesian coordinate $y$. **(D2)** Transformation of Fig. (D1) into the Cartesian coordinate $x$.

5. The PDE-law loss is satisfied only if $\hat{\hat{u}}_{\mathcal{B}} = \mathcal{B}(\hat{u})$, therefore yielding a PDE loss of the form

$$\mathcal{E}_{\text{PDE}} = \left\| \hat{\hat{u}}_{\mathcal{B}} - \mathcal{B}(\hat{u}) \right\|.$$

The proposed framework is depicted in Fig. 4. The boundary conditions (BC) are imposed in exactly the same manner as in the classical setting, with a PINN training loss taking the form $\mathcal{E}_{\text{PINN}} = \mathcal{E}_{\text{PDE}} + \mathcal{E}_{\text{BC}}$. It is, in fact, possible to capture boundary conditions using a coordinate-free formulation as well. For details, see M

In Fig. 14 (Appendix M) we compare solutions to the 2D Helmholtz equation on the unit square with the flat metric, obtained using (i) PINNs and (ii) the projection-based algorithm described above, where the PDE law is implicitly encoded by the fixed PCA model in coordinate-free space. For details, see Appendix M.

*Remark:* While the PDE constraint itself can be encoded as a relation (linear or nonlinear) between the coordinate-free basis elements in any input space, it *does not* inform us about what the necessary and sufficient initial and boundary conditions are to solve a PDE in a different geometry or dimension. For a relevant discussion in terms of well-posedness, see Bertalan et al. (2025)). This is *not* the case for time-evolution problems, where knowledge of the initial field is sufficient.

## 3.3 VECTOR PDEs

**Incompressible Navier Stokes:**

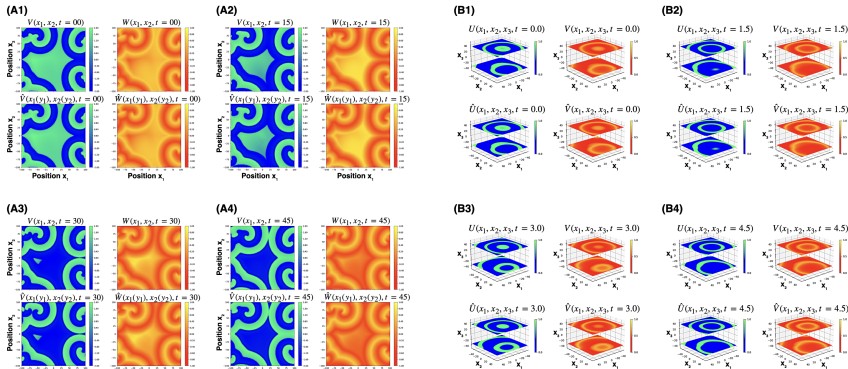

**Figure 3: The time evolution dynamics of FitzHugh-Nagumo and Barkley systems in higher-dimensional Euclidean spaces.** We study the FitzHugh-Nagumo system on a two-dimensional domain with periodic boundary conditions under various coordinate representations in **(A)**, and investigate the Barkley system in three-dimensional space subject to Neumann (no-flux) boundary conditions in **(B)**. For **(A)**, we present the time evolution of the true dynamics obtained by integrating Eq. (4) and compare it with the dynamics generated using the learned model at four different time frames: **(A1)** $t = 0$, **(A2)** $t = 15$, **(A3)** $t = 30$, **(A4)** $t = 45$. Each time frame consists of four two-dimensional plots: the top row shows the true dynamics, while the bottom row displays the neural network predictions; the left column corresponds to $V$-field and the right column to $W$-field of the FitzHugh-Nagumo system. For **(B)**, we present the time evolution of the true dynamics obtained by integrating Eq. (5) and compare it with the dynamics generated using the learned model at four different time frames: **(B1)** $t = 0$, **(B2)** $t = 1.5$, **(B3)** $t = 3.0$, **(B4)** $t = 4.5$. Every time frame consists of four three-dimensional plots (each illustrated with two $x_3$-slices, located at $x_3 = \pm 35$): the top row shows the true dynamics, while the bottom row displays the neural network predictions; the left column corresponds to $U$-field and the right column to $V$-field of the Barkley system.

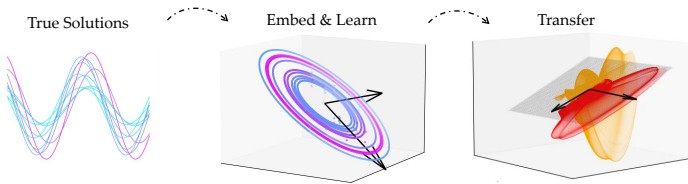

**Figure 4:** Implicit Coordinate-Free Representation: **Left:** Solutions of the 1-D Helmloltz problem (harmonic oscillator) in the original variables. **Middle:** Embedding of the solutions to $\mathcal{B}$, with black arrows depicting the two PCA modes obtained from the data. **Right:** Embedding of *one* prediction of a randomly initialized PINN in $\mathcal{B}$ (yellow), along with its projection (red) on the previously learned PCA submanifold . The PDE loss tests whether the yellow and green representations agree, i.e. if the PINN solution resides on the learned submanifold.

We apply our extended methodology to the 2D incompressible Navier-Stokes equations. We consider two well-studied settings, that of flow around a cylinder, and the Taylor-Green Vortex (Taylor & Green, 1937), illustrated in Figure 5. In the latter case, an analytical solution is known. The equation takes the form:

$$\frac{\partial v}{\partial t} = -(v \cdot \boldsymbol{\nabla})v - \frac{1}{\rho}\boldsymbol{\nabla}p + \frac{\eta}{\rho}\Delta v \tag{8}$$

$$\boldsymbol{\nabla} \cdot v = 0, \tag{9}$$

where $v = (v_1, v_2)$ is the velocity vector field, $p$ is the pressure, a scalar quantity, $\rho$ is the density which is assumed to be constant, and $\eta$ is the dynamic coefficient of the incompressible fluid, and is also constant. It is easy to see that this is an instance of the general form (3).

We will consider the problem of *learning* the momentum equation from the flow around a cylinder, and transferring it to the taylor-green vortex setting. Because of the simplified nature of the incompressible equations, we can

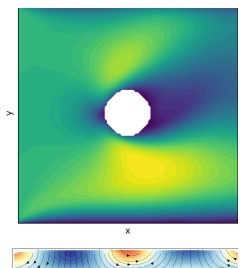

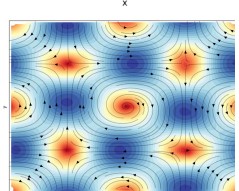

**Figure 5:** Magnitude of velocity field for simulated flow around cylinder (top) and integrated (learned, coordinate-free) Taylor-Green Vortex dynamics (bottom)

perform least-squares to identify the coefficients of the vector features (which play the role of the $f_i$ in (3)) instead of using a neural network. We note that the compressible Navier-Stokes would require the $f_i$ be nonlinear, at least as a function of density. We follow a familiar pattern: We compute the coordinate-free features of the Navier-Stokes solution in the cylinder setting, and perform least-squares to learn $\frac{\partial v}{\partial t}$ as a function of the spatial-derivative features. We then transfer the operator to the Taylor-Green Vortex setting, integrating using an explicit Euler method with a Poisson solve to handle the unknown pressure field. We compare against an initial condition for which the analytical solution is known. In these two examples, the topology of the domain is different. For details, see Appendix P.

## 4 DISCUSSION

We propose a framework that, for a general class of PDE systems, allows us to learn and represent the operator in a physically consistent, coordinate- and dimension-agnostic manner. The training can be performed using data from simple 1-D numerical simulations or experimental settings, yet it can generalize, approximating dynamics across dimensions, coordinate systems, and domain curvatures, even when varying boundary conditions. This is achieved *without* retraining, only by recomputing the invariant features in the target domain. Training can also be performed "fusing" available data from multiple dimensions or geometries. This flexibility makes our approach particularly valuable at the interface between experiments and engineering, as it enables seamless integration of experimental data from different geometry/dimension experiments into predictive models, allowing theoretical frameworks to adapt to complex real-world geometries.

Using ideas from any-dimensional machine learning, in this work, we focus on identification of second-order PDEs that are invariant under the full local symmetry group of Euclidean space (rotations and reflections, and translations) – which often emerge from long-wavelength physical effective field theories (Kaplan, 2005; Buchalla et al., 2014).In its current form, our framework handles only local differential operators. Nonlocal PDEs (e.g., Boltzmann) and more generally solution operators (mapping initial/boundary conditions to full PDE solutions) involve integral terms that are not encoded in the library, and may be domain-dependent; extending our framework to be compatible with this more general perspective is an important step in future work.

The next step is to apply this framework to *solving* PDEs in a dimension agnostic manner that do not explicitly require the use of integrators (here we use coordinate-free integrators). For example, by defining an any-dimensional formulation of neural operators or PINNs that allow us to transfer PDE solutions from one domain to another.

Our framework currently handles local operators;

Implicit to any-dimensional generalization is the underlying assumption of what it means to take a model from one space to another. In the case of graph neural networks assumptions that GNNs make to generalize to different graph sizes have been studied extensively in the graph transferability literature (Ruiz et al., 2020; Maskey et al., 2023; Velasco et al., 2024; Levin et al., 2025). For PDEs, the generalization to other domains is done via the intrinsic manifold operators (Laplacians, exterior derivatives, etc), but more work is needed to fully characterize what these methods are doing.

We expect this type of generalization to be broadly beneficial for the machine learning community, as PDEs play a central role not only in modeling of physical systems, but also optimization algorithms, sampling, and diffusion algorithms, which are fundamental to generative AI. A theoretically grounded approach to distilling information that can be applied to systems of arbitrary dimension could significantly boost efficiency for high-dimensional methods.

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

## A  Background on Differential Forms

The natural mathematical formalism to express coordinate-free operators on Riemannian manifolds is given by exterior calculus, which is built on differential forms. In this section, we give the relevant definitions and background.

A differential form is a mathematical object that generalizes scalars, vectors, and volumes, enabling integration and differentiation on manifolds in a coordinate-free and, to some extent, also in a dimension-free manner. A general $k$-form is a combination of $k$-differentials:

$$\Omega = \omega_{\mu_1\mu_2\ldots\mu_k} dx^{\mu_1} \wedge dx^{\mu_2} \wedge \ldots \wedge dx^{\mu_k} , \tag{10}$$

where $\omega_{\mu_1\mu_2\ldots\mu_k}$ are asymmetric tensor components, $\wedge$ denotes the wedge product which is anti-commutative, and note that we use the Einstein summation convention unless otherwise specified. On a Riemannian manifold, the contravariant version of these components can be obtained by raising the indices using the inverse metric tensor $g^{\mu\nu}$ (which is the inverse of the symmetric metric tensor $g_{\mu\nu}$, i.e. $g^{\mu\nu}g_{\mu\rho} = \delta^\nu_\rho$, where $\delta$ is the Kronecker delta tensor):

$$\omega^{\mu_1\mu_2\ldots\mu_k} = g^{\mu_1\nu_1} g^{\mu_2\nu_2} \ldots g^{\mu_k\nu_k} \omega_{\nu_1\nu_2\ldots\nu_k} . \tag{11}$$

The Hodge star is the operator that turns a $k$-form into a $(n-k)$-form:

$$\star\Omega = \frac{k!}{(n-k)!} \epsilon_{\mu_1\mu_2\ldots\mu_d} \omega^{\mu_1\mu_2\ldots\mu_k} dx^{\mu_{k+1}} \wedge dx^{\mu_{k+2}} \wedge \ldots \wedge dx^{\mu_n} , \tag{12}$$

in which $\epsilon_{\mu_1\mu_2...\mu_d}$ is the Levi-Civita tensor (exists on orientable manifolds). The exterior derivative $d\Omega$ is a $(k+1)$-form, which can be calculated with:

$$d\Omega = d\omega_{\mu_1\mu_2...\mu_k} \wedge dx^{\mu_1} \wedge dx^{\mu_2} \wedge ... \wedge dx^{\mu_{k-1}} \wedge dx^{\mu_k} , \tag{13}$$

where $d\omega_{\mu_1\mu_2...\mu_k}$ corresponds to the differential of the component $\omega_{\mu_1\mu_2...\mu_k}$, i.e.

$$d\omega_{\mu_1\mu_2...\mu_k} = \partial_{\mu_{k+1}}\omega_{\mu_1\mu_2...\mu_k} dx^{\mu_{k+1}} . \tag{14}$$

In a coordinate-explicit representation, given the metric structure $g$ defined on the manifold $\mathcal{M}$, the Laplacians and the inner products of gradients from Section 2 can be written as:

- The Laplacians:

$$\Delta\Psi_j = \star d \star d\Psi_j = |g|^{-1/2}\, \partial_\mu \left(|g|^{1/2}\, g^{\mu\nu}\partial_\nu\Psi_j\right) , \tag{15}$$

   where $j = 1, 2, 3, ...$ and $|g|$ is the determinant of the metric tensor $g_{\mu\nu}$. There are $N$ of them in total.

- The inner products of the gradients:

$$\langle\nabla\Psi_j, \nabla\Psi_k\rangle = \langle d\Psi_j, d\Psi_k\rangle \equiv \star(d\Psi_j \wedge \star d\Psi_k) = g^{\mu\nu}\partial_\mu\Psi_j\partial_\nu\Psi_k , \tag{16}$$

   where $j, k = 1, 2, 3, ...$ Since these products are symmetric, there are $N(N+1)/2$ distinct inner products in total.

# B    NUMERICAL SCHEME

To investigate systems of PDEs numerically, we need to specify the differentiation and integration schemes because they govern how the spatio-temporal continuous PDEs are approximated and then solved in a discrete form. The chosen methods influence the accuracy, stability, computational cost, and the appropriateness of the numerical solution.

## B.1    SPATIAL-DIFFERENTIATION

Here, the continuous spatial-derivatives in the PDEs are replaced by discrete approximations, i.e. central finite differences at the second-order accuracy. For simplicity, we consider the spatial-differentiation, such as $\partial_x, \partial_x^2$, of one-dimensional space scalar field, e.g. $M(x, t)$, $N(x, t)$. Generalizing to higher dimensions is straightforward, as each coordinate is treated independently.

For the first spatial-derivative, given that $\Delta x$ is the spacing between adjacent grid points, we use the following formula at each grid points:

$$\partial_x M(x, t) \overset{\text{num}}{\approx} \frac{M(x + \Delta x) - M(x - \Delta x)}{2\Delta x} .$$

For the second spatial-derivative, we apply:

$$\partial_x^2 M(x, t) \overset{\text{num}}{\approx} \frac{M(x + \Delta x) - 2M(x) + M(x - dx)}{\Delta x^2} .$$

Note that $\partial_x^2 M(x, t) \overset{\text{num}}{\approx} \partial_x [\partial_x M(x, t)]$ up to a small error of order $\mathcal{O}(\Delta x^2)$. When dealing with spatial derivatives inside other spatial derivatives, it can be beneficial to split them out rather than applying the derivatives consecutively before using the numerical approximations, e.g.

$$\partial_x [N(x, t)\partial_x M(x, t)] \rightarrow \partial_x N(x, t)\partial_x M(x, t) + N(x, t)\partial_x^2 M(x, t) \overset{\text{num}}{\approx} ... . \tag{17}$$

This splitting helps control error propagation, maintains higher accuracy, and also provides better handling of boundary conditions, preventing error accumulation near boundaries.

When considering a finite domain e.g. $x \in [-L_x, +L_x]$, the boundary conditions at the edges of the domain i.e. $x = \mp L_x$ dictate how we should modify Eq. (B.1) and Eq. (B.1) for points lying outside of the domain. For examples:

- Periodic boundary conditions:

$$M(\mp L_x \mp \Delta x) = M(\pm L_x) . \tag{18}$$

- Neumann (no-flux) boundary conditions:

$$M(\mp L_x \mp \Delta x) = M(\mp L_x) . \tag{19}$$

   We can see directly from this identification that $\partial_x M(x = \mp L_x) = 0$.

### B.2 TEMPORAL-INTEGRATION

Here, we use the forward Euler method to discretize the time derivative with spacing $\Delta t$:

$$\partial_t M(x,t) \overset{\text{num}}{\approx} \frac{M(x, t+\Delta t) - M(x,t)}{\Delta t} \, , \tag{20}$$

thus:

$$\partial_t M(x,t) = \mathcal{F}(t) \implies M(x, t+\Delta t) = M(x,t) + \mathcal{F}(t)\Delta t \, , \tag{21}$$

where $\mathcal{F}$ is a function that can depend not only on time position $t$, but also on space location, the fields value, and their spatial-derivatives e.g. the right-hand-side of the PDE. To numerically integrate $M(x,t)$, given the initial field value, we use Eq. (21) consecutively every timestep $t \to t + \Delta t$.

## C  EQUATIONS IN COORDINATE-FREE FORM

Here we give a coordinate-free expression for the PDEs from Section 3.1 using the formalism from Appendix A.

**FitzHugh-Nagumo System:**

$$\partial_t V(t) = \star\, d \star dV(t) + V(t) - \frac{1}{3}V^3(t) - W(t) \, ,$$
$$\partial_t W(t) = \varepsilon\left[V(t) + \beta - \gamma W(t)\right] \, , \tag{22}$$

**Barkley System:**

$$\partial_t U(t) = \star\, d \star dU(t) + \frac{1}{\varepsilon}U(t)\left[1 - U(t)\right]\left[U(t) - \frac{V(t)+b}{a}\right] \, ,$$
$$\partial_t V(t) = U(t) - V(t) \, , \tag{23}$$

**Modified Patlak-Keller-Segel System:**

$$\partial_t A(t) = \star\, d \star dA(t) - \left[\frac{A(t)}{1 + A(t)}\right]\left[\frac{B(t)}{1 + B(t)}\right] \, ,$$
$$\partial_t B(t) = \alpha B(t) + \star\, d \star d\left[D_B B(t)\right] - \star\, d \star d\left\{\chi[B(t)]\Phi[A(t)]\right\} \, , \tag{24}$$

## D  NUMERICAL RESULTS FOR TIME-EVOLUTION PDES

The learning of all PDE systems is conducted on data from a one-dimensional periodic domain $x \in [-L, +L]$. The data and neural network architectures are detailed in Appendix E. All simulations use evenly spaced grids for finite-difference computations in the specified coordinate system, and we verify that the numerical schemes are robust to space-time discretizations (see Appendix I). For notational convenience, we denote the fields obtained by numerically integrating the exact PDE as $\Psi_i$, and the predictions generated by the neural network models as $\hat{\Psi}_i$. The argument $\eta(\zeta)$ denotes that the computation was performed in $\zeta$-coordinates and has been push-forwarded into $\eta$-coordinates via a coordinate transformation. A uniform grid in $\zeta$-coordinates will in general correspond to a non-uniform grid in $\eta$-coordinates.

All quantitative comparisons between the "true" and learned dynamics are contained in Appendix K.

### D.1  ONE-DIMENSIONAL EUCLIDEAN SPACE

In this example, we learn the Patlak-Keller-Segel field evolution dynamics in a periodic one-dimensional space. We train a network with data in 1-D Cartesian coordinates $x$, and we test its performance under a change of coordinates $y$ in the same 1-D domain. Specifically $y$ is related to $x$ by the monotonic but nonlinear function:

$$x = f(y) \equiv x(y) = (y/2)\left[1 + (y/\lambda)^2\right] \, , \tag{25}$$

as illustrated in Fig.2A. We set $\lambda = L$. The inverse function $y = f^{-1}(x) \equiv y(x)$ is provided in Appendix G.1. The input pre-processing for the neural network (Fig. 1B, in $y$-coordinate) is detailed in Appendix G.2.

We test the trained model on a new initial condition $t = 0$ with uniform $A$ and localized peaks in $B$ (see Appendix I.1). The true dynamics, $A(x, t)$ and $B(x, t)$, are obtained by integrating Eq.(6) (Fig.2B1); the learned dynamics, $\hat{A}(x, t)$ and $\hat{B}(x, t)$, are generated by the neural network (Fig.2B2) are a good approximation. To test our coordinate-free generalization claims, we interpolate the initial field configurations to $y$-coordinates (see Appendix H), integrate the neural network model to get $\hat{A}(y, t)$ and $\hat{B}(y, t)$ (Fig.2D1), and then map the results back to $x$-space to arrive at $\hat{A}(x(y), t)$ and $\hat{B}(x(y), t)$ (Fig.2D2). We can also transform Fig. 1B1 to $y$-coordinates, obtaining $A(y(x), t)$ and $B(y(x), t)$ as shown in Fig. 2C. We observe that Fig. 2C and Fig. 2D1 agree, as well as Fig. 2A1 and Fig. 2D2, demonstrating that our machine learning method generalizes across coordinate systems.

### D.2 TWO- AND THREE-DIMENSIONAL EUCLIDEAN SPACE

In this section we consider the FitzHugh-Nagumo system and the Barkley system trained in 1-D cartesian data. We first add spatial-dimension to the FitzHugh-Nagumo system during test time. Using the trained models, we predict dynamics in both Cartesian $(x_1, x_2)$ and non-Cartesian $(y_1, y_2)$ coordinates, related by:

$$x_1 = f(y_1) \equiv x_1(y_1) \, , \ x_2 = f(y_2) \equiv x_2(y_2) \, , \tag{26}$$

where $f$ is as in Eq.(25) and we set $\lambda = L$. The domain is the square $(x_1, x_2) \in [-L, +L] \times [-L, +L]$ with periodic boundaries. An illustration of $(y_1, y_2)$ versus $(x_1, x_2)$ is in Appendix G.1; input pre-processing in $(y_1, y_2)$-coordinates for the neural network is detailed in Appendix G.3.

Spiral waves can emerge for certain instances of the FitzHugh-Nagumo system. We choose an initial field configuration at $t = 0$ where the spiral wave pattern is clearly visible (an explanation of how we do it is provided in Appendix I.2). In $(x_1, x_2)$-coordinates, the true dynamics are computed by integrating Eq.(4) (Fig. 3A1-4 top row, $V(x_1, x_2, t)$ and $W(x_1, x_2, t)$). In $(y_1, y_2)$-coordinates, the learned model is integrated from the extrapolated initial condition (see Appendix H for details), and the results are mapped back to $(x_1, x_2)$ (Fig. 3A1-4 bottom row, $\hat{V}(x_1(y_1), x_2(y_2), t)$ and $\hat{W}(x_1(y_1), x_2(y_2), t)$). We observe that the predictions agree, suggesting that our machine learning method not only generalizes across coordinate systems, but also it extends across dimensions. The correct predictions are made in two dimensions, even though the model was trained in one dimension, indicating that the approach is dimension-free.

We now consider the Barkley system, and we show that the learning in 1-D can generalize to **three spatial dimensions** with Neumann (no-flux) boundary conditions on a cubic Cartesian domain $(x_1, x_2, x_3) \in [-L, +L] \times [-L, +L] \times [-L, +L] \equiv \mathcal{C}$, i.e. for all fields $\Psi_i$.

$$\star d\Psi_i \big|_{\partial \mathcal{C}} = 0 \, , \tag{27}$$

which is equivalent to stating that the normal derivative $\hat{e}_\perp \cdot \vec{\nabla} \Psi_i$ vanishes on $\partial \mathcal{C}$ (where $\hat{e}_\perp$ is the unit-vector normal to $\partial \mathcal{C}$ locally). These boundary conditions differ from those used during training and in previous cases.

We initialize the Barkley system with a field configuration that clearly displays scroll wave patterns (Appendix I.3). Fig. 3B1–4 shows that the neural network predictions, $\hat{U}(x_1(y_1), x_2(y_2), x_3(y_3), t)$ and $\hat{V}(x_1(y_1), x_2(y_2), x_3(y_3), t)$, closely match the true dynamics, $U(x_1, x_2, x_3, t)$ and $V(x_1, x_2, x_3, t)$, as governed by Eq. (5), demonstrating the adaptability of our dimension-free model to diverse boundary conditions.

### D.3 TWO-DIMENSIONAL CURVED SURFACE EMBEDDED IN 3-DIMENSIONAL EUCLIDEAN SPACE

In addition to increasing the number of the spatial dimensions and modifying the boundary conditions, we now *bend the space* (intrinsically). For this case, we explore the dynamics of the FitzHugh–Nagumo fields on a two-dimensional domain (similar to Fig. 3A but not flat) embedded in a spherical surface — a manifold with constant positive intrinsic curvature. We consider two

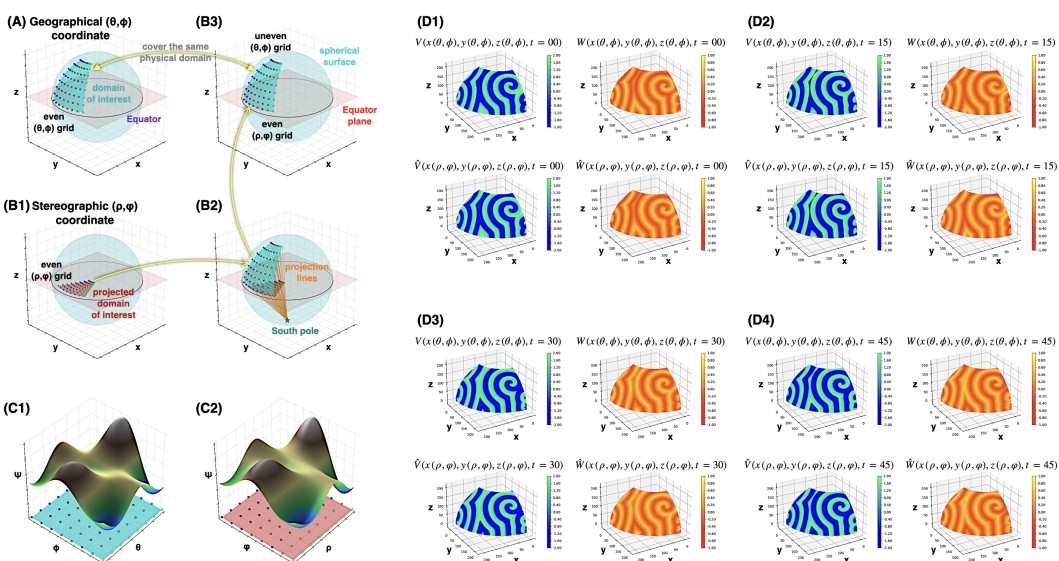

**Figure 6: The time evolution of the FitzHugh–Nagumo system on a two-dimensional intrinsically curved surface.** We illustrate the geographic and stereographic coordinate representations in **(A–C)**, and in **(D)** compare the true dynamics obtained by integrating Eq. (22) in geographic coordinates with the learned dynamics generated in stereographic coordinates. In **(A–B)**, we embed the spherical surface in three-dimensional Cartesian space $(x, y, z)$, showing the sphere in cyan, the equatorial plane in red, and their intersection—the equator—in purple: **(A)** shows a uniform grid in geographic coordinates $(\theta, \phi)$; **(B)** shows a uniform grid in stereographic coordinates $(\rho, \varphi)$—on the plane in **(B1)**, projected onto the sphere in **(B2)**, resulting in a non-uniform grid in $(\theta, \phi)$ as shown in **(B3)**. **(C1)** shows a field $\Psi$ defined over $(\theta, \phi)$, and **(C2)** shows the same field over $(\rho, \varphi)$. In **(D)**, we present field configurations at four time frames: **(D1)** $t = 0$, **(D2)** $t = 15$, **(D3)** $t = 30$, **(D4)** $t = 45$. Each time frame consists of four three-dimensional plots in embedded $(x, y, z)$-space: the top row shows the true dynamics, the bottom row shows the neural network predictions; the left column displays the $V$-field and the right column the $W$-field of the FitzHugh-Nagumo system.

coordinate representations of this curved space — the geographic $(\theta, \phi)$ and the stereographic $(\rho, \varphi)$ systems — as illustrated in Fig. 6A-C (See Appendix J for more details). The transformation between these coordinate systems is given by:

$$\rho = R \tan\left(\phi/2\right) \ , \ \varphi = \phi \ , \tag{28}$$

where $R$ is the radius of the sphere. These coordinates can be further related to the Cartesian coordinates $(x, y, z)$ of the ambient three-dimensional Euclidean embedding space via:

$$x = R \sin\theta \sin\phi \ , \ y = R \sin\theta \cos\phi \ , \ z = R \cos\theta \ . \tag{29}$$

The spatial domain will be constrained between the radial values $\rho_{\min}$ and $\rho_{\max}$, and the angular values $\varphi_{\min}$ and $\varphi_{\max}$, which can be determined from $\theta_{\min}, \theta_{\max}, \phi_{\min}, \phi_{\max}$ (Appendix I.4), We impose heterogeneous boundary conditions: the Neumann (no-flux) conditions on the latitude boundaries and the periodic conditions on the longitude boundaries.

We initialize the system with a clearly defined spiral pattern for the fields $V$ and $W$ (defined in $(\theta, \phi)$-coordinates, see Appendix I.4). The true dynamics — $V(x(\theta, \phi), y(\theta, \phi), z(\theta, \phi), t)$ and $W(x(\theta, \phi), y(\theta, \phi), z(\theta, \phi), t)$ — are obtained by integrating Eq.(4) in $(\theta, \phi)$-coordinates and then mapped to the embedding space $(x, y, z)$, as illustrated in the top row of Fig.6D1–4. For the stereographic $(\rho, \varphi)$-coordinates, we extrapolate the initial condition (Appendix H), integrate the neural network learned dynamics, and transform the results to $(x, y, z)$ to obtain $\hat{V}(x(\rho, \varphi), y(\rho, \varphi), z(\rho, \varphi), t)$ and $\hat{W}(x(\rho, \varphi), y(\rho, \varphi), z(\rho, \varphi), t)$, shown in the bottom row of Fig.6D1–4. We observe a remarkable agreement (quantified in Appendix K) between both predictions, illustrating that our machine learning method not only generalizes across coordinate systems, extends across dimensions, and adapts to different boundary conditions, but also remains robust under curvatures.

### D.4 TWO-DIMENSIONAL FLAT SURFACE WITH NON-TRIVIAL TOPOLOGY

Finally, we *change the topology* of the environment. In essence, this is straightforward, since only the boundary specification needs to be modified. For example, let us study the FitzHugh–Nagumo system on a flat annulus in the polar coordinates $(\rho, \theta)$, which can be related to the Cartesian coordinates $(x, y)$ via:

$$x = \rho \cos\theta \ , \ y = \rho \sin\theta \ , \tag{30}$$

where the region of interests is confined within $\rho \in [\rho_{\min}, \rho_{\max}]$, where we impose fixed-value Dirichlet boundary conditions.

Similar to what has been done in Section D.2, we choose an initial field configuration at $t = 0$ where the spiral wave pattern is clearly visible (an explanation of how we do it is provided in Appendix I.5). The true dynamics are computed by integrating Eq.(4) (Fig. 7A-D top row, $V(\rho, \theta, t)$ and $W(\rho, \theta, t)$). The learned model is integrated from the very same initial condition (Fig. 7A-D bottom row, $\hat{V}(\rho, \theta, t)$ and $\hat{W}(\rho, \theta, t)$). Both of these are mapped back to the $(x, y)$ coordinates, i.e. $V(x(\rho, \theta), y(\rho, \theta), t)$, $W(x(\rho, \theta), y(\rho, \theta), t)$, $\hat{V}(x(\rho, \theta), y(\rho, \theta), t)$, and $\hat{W}(x(\rho, \theta), y(\rho, \theta), t)$, for illustration purpose. We observe that the predictions agree, suggesting that our machine learning method not only generalizes across coordinate and intrinsically curved systems, but also applicable to non-trivial topologies. In other words, our "spatially liberated" approach is topology-free.

### E TRAINING THE NEURAL NETWORKS

We describe the training procedure and also the generation of training data within a one-dimensional periodic space equipped with a Cartesian coordinate where we impose periodic boundary conditions (Appendix B), i.e.

$$x \in \mathcal{S} = [-L, +L] \ , \ x \sim x + L \ , \tag{31}$$

and its use in training spatially liberated neural networks for each system as sketched in Fig. 1A.

For each of the systems (FitzHugh-Nagumo model, Barkley model, and Patlak-Keller-Segel model), we aim to train a neural network with an MLP architecture, consisting of $n_{lay} = 2$ hidden layers, each with $n_{neu}$ neurons (to be specified). This network takes as input seven scalar quantities (0-forms) which are invariant features as listed in Eq. (1):

$$\Psi_1 \ , \ \langle d\Psi_1, d\Psi_1 \rangle \ , \ \star d \star d\Psi_1 \ , \ \Psi_2 \ , \ \langle d\Psi_2, d\Psi_2 \rangle \ , \ \star d \star d\Psi_2 \ , \ \langle d\Psi_1, d\Psi_2 \rangle \tag{32}$$

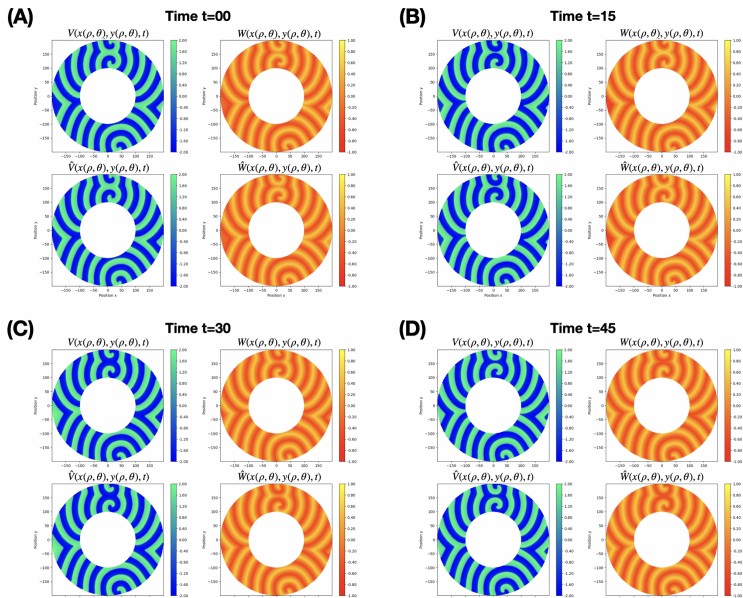

**Figure 7: The time evolution of the FitzHugh–Nagumo system on a two-dimension flat surface with an annulus topology.** We present the time evolution of the true dynamics obtained by integrating Eq. (4) and compare it with the dynamics generated using the learned model at four different time frames: **(A)** $t = 0$, **(B)** $t = 15$, **(C)** $t = 30$, **(D)** $t = 45$. Each time frame consists of four two-dimensional plots: the top row shows the true dynamics, while the bottom row displays the neural network predictions; the left column corresponds to $V$-field and the right column to $W$-field of the FitzHugh-Nagumo system.

to predict two changing rates of the dynamical fields

$$\partial_t \Psi_1 \ , \ \partial_t \Psi_2$$

at each location $x$ in space and point $t$ in time; see Fig. 1A. For the FitzHugh-Nagumo system, we set $n_{\text{neu}} = 64$, with the fields identified as $(\Psi_1, \Psi_2) = (V, W)$. In the Barkley system, we likewise use $n_{\text{neu}} = 64$, where $(\Psi_1, \Psi_2) = (U, V)$. With the Patlak-Keller-Segel system, we consider $n_{\text{neu}} = 32$, in which $(\Psi_1, \Psi_2) = (A, B)$.

The training data for each case is first simulated and then subsampled for training as described below: Section E.1 for the FitzHugh-Nagumo system, Section E.2 for the Barkley system, and Section E.3 for the modified Patlak-Keller-Segel system. For all simulations in this work, we have checked that the numerical computations are robust to space-time discretizations.

All computations in this work were performed on CPU-based machines. Each experiment was trained in under one hour using an internal compute node equipped with an Intel Core i7-7700 processor (3.60 GHz), 128 GB RAM, and local SSD storage. No GPUs or cloud services were used.

### E.1 FitzHugh-Nagumo System Training

We generate numerical simulation data of the FitzHugh-Nagumo system on a one-dimensional domain specified in Eq. (31), using $L = 10^2$. The spatiotemporal discretization is set to $(\Delta t, \Delta x) = (10^{-1}, 1)$, and for every numerical simulation of Eq. (22) we generate the time progression consecutively at each timestep $\Delta t$, using forward Euler method Eq. (21), for $t \in [0, T]$ where $T = 10^3$. All training data is obtained from a single simulation, in which the initial field configuration — consisting of combinations of a few localized configurations — is given in Appendix F.1. Then, we randomly-selected a subset of the simulation data (i.e. 10%) to create the training dataset for neural network training. This reduces the training data size but still ensures a fine time-increment, minimizing error propagation and improving long-term prediction accuracy of the trained model. The neural network model is optimized using the least squares deviation criterion with the Adam optimizer Kingma (2014), set to a learning rate of $5 \times 10^{-4}$, and trained for $2 \times 10^3$ epochs.

## E.2 BARKLEY SYSTEM TRAINING

We create numerical simulation data on the one-dimensional domain specified in Eq. (31) for Barkley system, using $L = 5 \times 10^1$. The spatiotemporal discretization is set to $(\Delta t, \Delta x) = (10^{-2}, 5 \times 10^{-1})$, and for every numerical simulation of Eq. (23) we generate the time progression consecutively at each timestep $\Delta t$, using the forward Euler method Eq. (21), for $t \in [0, T]$ where $T = 2 \times 10^2$. All training data is obtained from a single simulation, in which the initial field configuration — consisting of combinations of a few localized configurations — is given in Appendix F.2. Then, we randomly-selected a subset of the simulation data (i.e. $5\%$) to create the training dataset for neural network training. This reduces the training data size but still ensures a fine time-increment, minimizing error propagation and improving long-term prediction accuracy of the trained model. The neural network model is optimized using the least squares deviation criterion with the Adam optimizer, set to a learning rate of $5 \times 10^{-4}$, and trained for $2 \times 10^3$ epochs.

## E.3 PATLAK-KELLER-SEGEL SYSTEM TRAINING

We produce numerical simulation data for the Palak-Keller-Segel system on a one-dimensional domain specified in Eq. (31), using $L = 10^2$. The spatiotemporal discretization is set to $(\Delta t, \Delta x) = (10^{-1}, 1$, and for every numerical simulation of Eq. (24) we generate the time progression consecutively at each timestep $\Delta t$, using forward Euler method Eq. (21), for $t \in [0, T]$ where $T = 5 \times 10^2$. The training data is derived from multiple simulations showcasing traveling wave collisions and diffusion. The initial field configurations used to create these simulations are uniform for $A$ and consist of combinations of a few localized configurations for $B$, as provided in Appendix F.3. Our choice of domain dimensions and initial field configurations are motivated by their relevance to real experimental settings using microfluidic devices. Then, we randomly-selected a subset of the simulation data (i.e. $2\%$) to create the training dataset for neural network training. This reduces the training data size but still ensures a fine time-increment, minimizing error propagation and improving long-term prediction accuracy of the trained model. The neural network model is optimized using the least squares deviation criterion with the Adam optimizer, set to a learning rate of $5 \times 10^{-4}$, and trained for $2 \times 10^3$ epochs.

For notational convenience, we redefine $A$, $B$ to represent the normalized quantities $A/A_{\max}$, $B/B_{\max}$ respectively. This convention applies from this point forward unless specified otherwise. Note that $A_{\max} = 30$ and $B_{\max} = 1$.

# F INITIAL FIELD CONFIGURATIONS FOR TRAINING

Here we list the initial field configurations used to generate the training data for the FitzHugh-Nagumo, the Barkley, and the Patlak-Keller-Segel models, as explained in Appendix E.

## F.1 FITZHUGH-NAGUMO SYSTEM

The initial field configurations are given by:

$$
\begin{aligned}
V(x,0) &= V_{01} \exp\left[-\frac{(x - x_{V1})^2}{2\sigma_{xV1}}\right] + V_{02} \exp\left[-\frac{(x - x_{V2})^2}{2\sigma_{xV2}}\right] , \\
W(x,0) &= W_{01} \exp\left[-\frac{(x - x_{W1})^2}{2\sigma_{xW1}}\right] ,
\end{aligned}
\tag{33}
$$

where:

$$
\begin{aligned}
(V_{01}, x_{V1}, \sigma_{xV1}) &= (+2, +40, 10) , \\
(V_{02}, x_{V2}, \sigma_{xV2}) &= (-2, -40, 10) , \\
(W_{01}, x_{W1}, \sigma_{xW1}) &= (+1, 0, 10) .
\end{aligned}
\tag{34}
$$

## F.2  BARKLEY SYSTEM

The initial field configurations are given by:

$$U(x,0) = U_{01} \exp\left[-\frac{(x-x_{U1})^2}{2\sigma_{xU1}}\right]$$

$$+ U_{02} \exp\left[-\frac{(x-x_{U2})^2}{2\sigma_{xU2}}\right] + U_{03} \exp\left[-\frac{(x-x_{U3})^2}{2\sigma_{xU3}}\right] ,$$

$$V(x,0) = V_{01} \exp\left[-\frac{(x-x_{V1})^2}{2\sigma_{xV1}}\right]$$

$$+ V_{02} \exp\left[-\frac{(x-x_{V2})^2}{2\sigma_{xV2}}\right] + V_{03} \exp\left[-\frac{(x-x_{V3})^2}{2\sigma_{xV3}}\right] ,$$

(35)

where:

$$(U_{01}, x_{U1}, \sigma_{xU1}) = (0.9, +22, 4) ,$$
$$(U_{02}, x_{U2}, \sigma_{xU2}) = (0.5, -8, 4) ,$$
$$(U_{03}, x_{U3}, \sigma_{xU3}) = (0.8, -28, 2) ,$$
$$(V_{01}, x_{V1}, \sigma_{xV1}) = (0.5, 0, 4) ,$$
$$(V_{02}, x_{V2}, \sigma_{xV2}) = (0.9, 20, 2) ,$$
$$(V_{03}, x_{V3}, \sigma_{xV3}) = (0.7, -24, 2) .$$

(36)

## F.3  PATLAK-KELLER-SEGEL SYSTEM

Here we consider an active matter system of many chemotactic bacteria that navigate in response to chemo-attractants, which they simultaneously deplete through their own consumption Phan et al. (2024). Two important phenomena emerge from bacterial collective chemotaxis: the formation of traveling bands of bacteria and the collisions of those bands, which then diffuse Phan et al. (2020). To generate a series of *in silico* experiments illustrating both phenomena, we begin with simple initial conditions: the Asp is uniform, and bacteria are inoculated at two ports, each modeled by a Gaussian distribution. We consider four different initial profiles for Asp concentration and four times three distinct initial profiles for the bacterial field, i.e.

$$A(x,0) = A_0 ,$$

$$B(x,0) = B_1 \exp\left[-\frac{(x-x_{B1})^2}{2\sigma_{B1}}\right] + B_2 \exp\left[-\frac{(x-x_{B2})^2}{2\sigma_{B2}}\right] ,$$

(37)

where:

$$x_{B1} = -50 , \ x_{B2} = +50 , \ \sigma_{B1} = \sigma_{B2} = 5 ,$$
$$A_0 \in A_{\max} \times \{0, 1/4, 1/2, 3/4, 1\} ,$$
$$B_1 \in B_{\max} \times \{1/3, 2/3, 1\} , \ B_2 \in B_{\max} \times \{0, 1/3, 2/3, 1\} .$$

(38)

For notational convenience, we redefine $A$, $B$ to represent the normalized quantities $A/A_{\max}$, $B/B_{\max}$ respectively. This convention applies from this point forward unless specified otherwise. Note that $A_{\max} = 30$ and $B_{\max} = 1$. We have chosen the parameters so that these experiments are implementable within microfluidic environment.

## G  CARTESIAN AND NON-CARTESIAN COORDINATES

Here we show the relation between different choices of coordinate systems used in Section D.1 and Section D.2, from the mapping between locations to the calculation of relevant differential forms.

### G.1  INVERSE FUNCTION AND ILLUSTRATION OF $y$ VS $x$

The inverse function, which helps calculating $y$ from $x$, is given by:

$$y = f^{-1}(x) = \lambda \left[\frac{\Theta(x)}{\sqrt[3]{2} \cdot 3^{2/3}} - \frac{\sqrt[3]{\frac{2}{3}}}{\Theta(x)}\right] ,$$

(39)

where

$$\Theta(x) = \sqrt[3]{\sqrt{3}\sqrt{27\left(\frac{2x}{\lambda}\right)^2 + 4} + 9\left(\frac{2x}{\lambda}\right)}\,.$$

The transformations between Cartesian and non-Cartesian coordinates in different spaces are presented in Fig. 8.

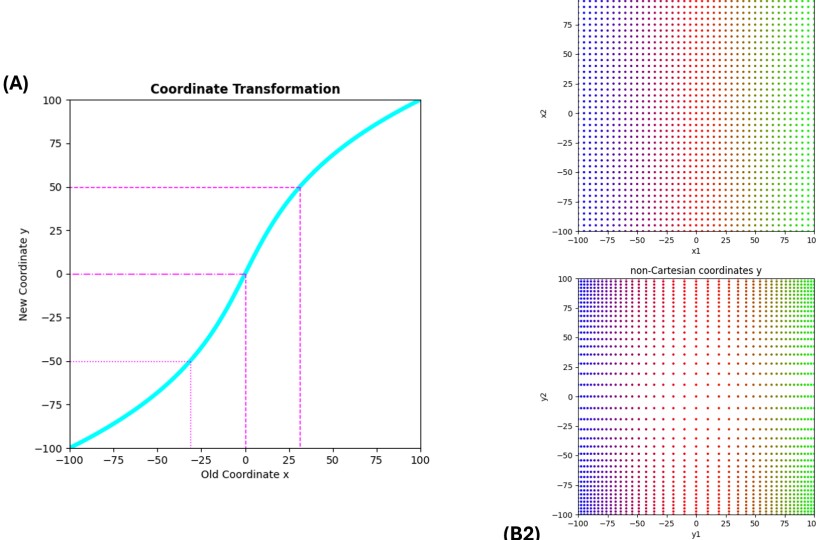

Figure 8: **The Cartesian and non-Cartesian coordinate systems in one- and two-dimensional space.** Here we use Eq. (25) and Eq. (26) with $\lambda = L = 10^2$. **(A)** One-dimensional space, conversion between $x$ and $y$. **(B)** Two-dimensional space, conversion between $(x_1, x_2)$ and $(y_1, y_2)$. The color transitions linearly with $x_1$, following the Python Matplotlib brg colormap Hunter (2007), shifting from blue to red to green.

### G.2 CALCULATION OF DIFFERENTIAL-FORMS IN ONE-DIMENSIONAL EUCLIDEAN SPACE

Denote $' = d/dy$ when taking the derivative of the function $f(y)$. The differential-form inputs (besides $\Psi_1$ and $\Psi_2$) used in the neural network as illustrated in Fig. 1B for the one-dimensional non-Cartesian case (Section D.1) can be calculated as follows:

- The Laplacians:

$$\star d \star d\Psi_j(t) = \frac{1}{|f'(y)|}\left\{\frac{|f'(y)|}{[f'(y)]^2}\right\}' \partial_y \Psi_j(y,t) + \frac{1}{[f'(y)]^2}\partial_y^2 \Psi_j(y,t)\,, \qquad (40)$$

where $j = 1, 2$. There are two of such inputs.

- The inner products:

$$\langle d\Psi_j(t), d\Psi_k(t)\rangle = \left[\frac{1}{f'(y)}\partial_y \Psi_j(y,t)\right]\left[\frac{1}{f'(y)}\partial_y \Psi_k(y,t)\right]\,, \qquad (41)$$

where $j, k = 1, 2$. There are three of such inputs, corresponding to three distinct inner products.

### G.3 CALCULATION OF DIFFERENTIAL-FORMS IN TWO-DIMENSIONAL EUCLIDEAN SPACE

The differential-form inputs (besides $\Psi_1$ and $\Psi_2$) used in the neural network as illustrated in Fig. 1B for the two-dimensional non-Cartesian case (Section D.2) can be calculated as follows:

- The Laplacians:

$$\star d \star d\Psi_j(t) = \frac{1}{|f'(y_1)f'(y_2)|^{1/2}} \partial_{y_1} \left\{ \frac{|f'(y_1)f'(y_2)|^{1/2}}{[f'(y_1)]^2} \right\} \partial_{y_1} \Psi_j(y_1, y_2, t)$$
$$+ \frac{1}{|f'(y_1)f'(y_2)|^{1/2}} \partial_{y_2} \left\{ \frac{|f'(y_1)f'(y_2)|^{1/2}}{[f'(y_2)]^2} \right\} \partial_{y_2} \Psi_j(y_1, y_2, t) \qquad (42)$$
$$+ \frac{1}{[f'(y_1)]^2} \partial_{y_1}^2 \Psi_j(y_1, y_2, t) + \frac{1}{[f'(y_2)]^2} \partial_{y_2}^2 \Psi_j(y_1, y_2, t) \,,$$

where $j = 1, 2$. There are two of such inputs.
- The inner products:

$$\langle d\Psi_j(t), d\Psi_k(t) \rangle = \left[ \frac{1}{f'(y_1)} \partial_{y_1} \Psi_j(y_1, y_2, t) \right] \left[ \frac{1}{f'(y_1)} \partial_{y_1} \Psi_k(y_1, y_2, t) \right]$$
$$+ \left[ \frac{1}{f'(y_2)} \partial_{y_2} \Psi_j(y_1, y_2, t) \right] \left[ \frac{1}{f'(y_2)} \partial_{y_2} \Psi_k(y_1, y_2, t) \right] \,, \qquad (43)$$

where $j, k = 1, 2$. There are three of such inputs, corresponding to three distinct inner products.

## H    COMPARISON BETWEEN DIFFERENT COORDINATE SYSTEMS

We illustrate the comparison between coordinate systems — using 1D Cartesian and non-Cartesian coordinates as an example — which naturally extends to higher dimensions and curved spaces. For a given dynamical system, we initialize a condition (distinct from the training set) in $x$-space with uniform $\Delta x$ discretization. We then compute its evolution via both the true PDE and the neural network model from Section E, showing strong agreement. The corresponding initial condition in $y$-space (with uniform $\Delta y$) is evolved using the neural network, then mapped back to $x$-space via Eq. (25) for direct comparison. These steps are summarized in Fig. 9. For a coordinate-free model, all these generated datasets should be consistent, which we confirm through our results.

## I    INITIAL FIELD CONFIGURATIONS FOR TESTING

Here we list the initial field configurations used for testing in Section D.1 (see Appendix I.1), Section D.2 (see Appendix I.2), Section D.2 (see Appendix I.3), and Section D.3 (see Appendix I.4).

### I.1    ONE-DIMENSIONAL EUCLIDEAN SPACE

In this case (Section D.1), we work with the Patlak-Keller-Segel system. The initial field configurations The initial field configurations used in testing follow similar structure to Eq. (38), with:

$$x_{B1} = 10 \,, \; x_{B2} = -60 \,, \; \sigma_{B1} = 5 \,, \; \sigma_{B2} = 6 \,,$$
$$A_0 = 5/6 \,, \; B_1 = 0.7 \,, \; B_2 = 0.4 \,. \qquad (44)$$

We consider the same periodic space as in Appendix E for Patlak-Keller-Segel system, i.e. $x \in [-L, +L]$ with $L = 10^2$ and a uniform grid of $\Delta x = 1$. The transformation from this Cartesian coordinate $x$ to a non-Cartesian one $y$ follows Eq. (25). We use $y \in [-L, +L]$ with a uniform grid and a lattice spacing of $\Delta y = 1$ when predicting the field evolution dynamics by integrating the trained neural network model from Section F.3 in this coordinate system.

We start with an initial field configuration that has not been used to generate the training data in Appendix E. The field $A$ is uniform everywhere and the field $B$ are combinations of localized configurations, as provided in Eq. (44). We create the field evolution dynamics using the timestep $\Delta t = 10^{-1}$ for $t \in [0, 5 \times 10^2]$ by numerically integrating the PDEs Eq. (6).

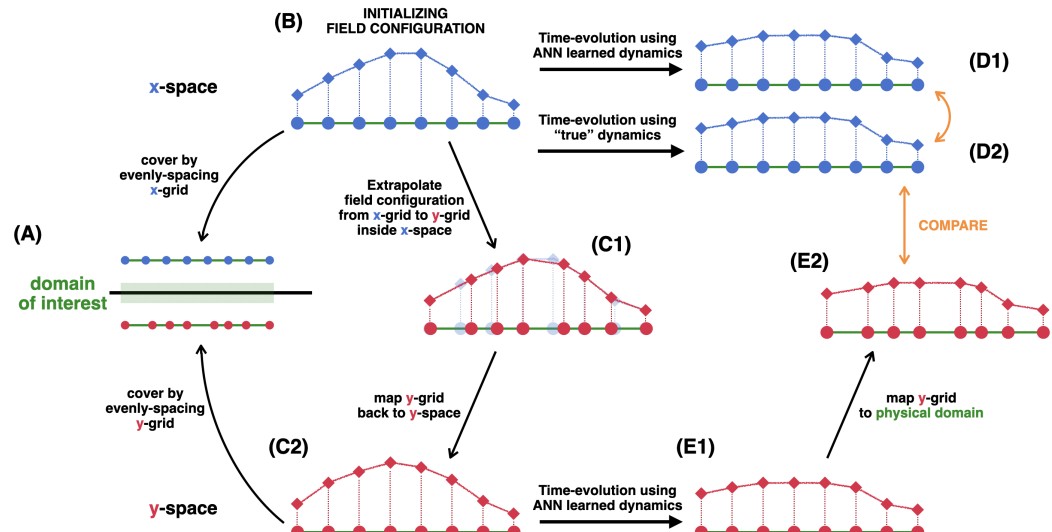

**Figure 9: Testing coordinate-free learning in one-dimensional space.** We study the time-evolution in different coordinates, using both "true" computed dynamics and ANN learned dynamics. **(A)** The physical domain of interest can be described (parametrized) by either the $x$-coordinate or the $y$-coordinate. An evenly-spaced $x$-grid is not equivalent to an evenly-spaced $y$-grid. **(B)** We start with an initial field configuration in $x$-space. **(C1,C2)** We extrapolate the values of the initial field configuration from what it looks like in $x$-space to its estimation in $y$-space. **(D)** In $x$-space, we compute the time-evolution of the field in two different ways: using the "true dynamics" **(D1)** and using the neural network learned dynamics **(D2)**. We then can compare them. **(E1)** In $y$-space, we compute the time-evolution of the field using the neural network learned dynamics. **(E2)** We map the field back to physical space, showing what it looks like in $x$-space, then compare with **(D)**.

### I.2  Two-Dimensional Euclidean Space

In this case (Section D.2), we work with the FitzHugh-Nagumo system. We start with the field configuration given by:

$$V(x_1, x_2) = V_{01} \exp\left[-\frac{(x_1 - x_{1V1})^2}{2\sigma_{x1V1}} - \frac{(x_2 - x_{2V1})^2}{2\sigma_{x2V1}}\right]$$

$$+ V_{02} \exp\left[-\frac{(x_1 - x_{1V2})^2}{2\sigma_{x1V2}} - \frac{(x_2 - x_{2V2})^2}{2\sigma_{x2V2}}\right], \tag{45}$$

$$W(x_1, x_2) = W_{01} \exp\left[-\frac{(x_1 - x_{1W1})^2}{2\sigma_{x1W1}} - \frac{(x_2 - x_{2W1})^2}{2\sigma_{x2W1}}\right],$$

where:

$$(V_{01}, x_{1V1}, x_{2V1}, \sigma_{x1V1}, \sigma_{x2V1}) = (+1, 0, 0, 15, 10),$$
$$(V_{02}, x_{1V2}, x_{2V2}, \sigma_{x1V2}, \sigma_{x2V2}) = (+1, 0, +40, 8, 8), \tag{46}$$
$$(W_{01}, x_{1W1}, x_{2W1}, \sigma_{x1W1}, \sigma_{x2W1}) = (+1, +10, 0, 10, 25).$$

Both $V$ and $W$ are combinations of localized configurations. Then, we numerically integrate the PDEs Eq. (4) for the total amount of time $\Delta T = 8 \times 10^2$ (so that the system has reached the stage of steady spiral wave generation) from the above configurations. We use the final-time configuration as the initial field configuration for the study in Section D.2.

Consider the periodic spatial domain $(x_1, x_2) \in [-L, +L] \times [-L, +L]$ with $L = 10^2$ and uniform grid of lattice-spacing $(\Delta x_1, \Delta x_2) = (1, 1)$. The transformation from these Cartesian coordinates $(x_1, x_2)$ to non-Cartesian ones $(y_1, y_2)$ is as described Eq. (26). We use $(y_1, y_2) \in [-L, +L] \times [-L, +L]$ with a uniform grid and a lattice-spacing of $(\Delta y_1, \Delta y_2) = 1$ to describe the same space in this new coordinate system.

Since $(x_1, x_2)$-grid and $(y_1, y_2)$-grid do not represent identical set of physical points, we need to extrapolate the configuration Eq. (46) to $(y_1, y_2)$-coordinate. We generate the field evolution dynamics using the timestep $\Delta t = 10^{-1}$ for $t \in [0, 10^2]$, in different coordinate systems.

## I.3 THREE-DIMENSIONAL EUCLIDEAN SPACE

In this case (Section D.2), we work with the Barkley system. We start with the field configuration given by:

$$
\begin{aligned}
U(x_1, x_2, x_2) = {} & U_{01} \exp\left[ -\frac{(x_1 - x_{1U1})^2}{2\sigma_{x1U1}} - \frac{(x_2 - x_{2U1})^2}{2\sigma_{x2U1}} - \frac{(x_3 - x_{3U1})^2}{2\sigma_{x3U1}} \right] \\
& + U_{02} \exp\left[ -\frac{(x_1 - x_{1U2})^2}{2\sigma_{x1U2}} - \frac{(x_2 - x_{2U2})^2}{2\sigma_{x2U2}} - \frac{(x_3 - x_{3U2})^2}{2\sigma_{x3U2}} \right] , \\
V(x_1, x_2, x_3) = {} & V_{01} \exp\left[ -\frac{(x_1 - x_{1V1})^2}{2\sigma_{x1V1}} - \frac{(x_2 - x_{2V1})^2}{2\sigma_{x2V1}} - \frac{(x_3 - x_{3V1})^2}{2\sigma_{x3V1}} \right] \\
& + V_{02} \exp\left[ -\frac{(x_1 - x_{1V2})^2}{2\sigma_{x1V2}} - \frac{(x_2 - x_{2V2})^2}{2\sigma_{x2V2}} - \frac{(x_3 - x_{3V2})^2}{2\sigma_{x3V2}} \right] ,
\end{aligned}
\tag{47}
$$

where:

$$
\begin{aligned}
(U_{01}, x_{1U1}, x_{2U1}, x_{3U1}, \sigma_{x1U1}, \sigma_{x2U1}, \sigma_{x3U1}) &= (0.5, 0, 6, 0, 5, 6, 5) , \\
(U_{02}, x_{1U2}, x_{2U2}, x_{3U2}, \sigma_{x1U2}, \sigma_{x2U2}, \sigma_{x3U2}) &= (0.5, 0, 12, 6, 4, 5, 4) , \\
(V_{01}, x_{1V1}, x_{2V1}, x_{3V1}, \sigma_{x1V1}, \sigma_{x2V1}, \sigma_{x3V1}) &= (0.8, 5, 6, 0, 5, 4, 4) , \\
(V_{02}, x_{1V2}, x_{2V2}, x_{3V2}, \sigma_{x1V2}, \sigma_{x2V2}, \sigma_{x3V2}) &= (0.7, -7, 6, -7, 4, 4, 6) .
\end{aligned}
\tag{48}
$$

Both $U$ and $V$ are combinations of localized configurations. Then, we numerically integrate the PDEs Eq. (5) for the total amount of time $\Delta T = 2 \times 10^1$ (so that the system has reached the stage of steady scroll wave generation) from the above configurations. We use the final-time configuration as the initial field configuration for the study in Section D.2.

We consider the confined space $(x_1, x_2, x_3) \in [-L, +L] \times [-L, +L] \times [-L, +L]$ with $L = 5 \times 10^1$, covered by a uniform grid of lattice-spacing $(\Delta x_1, \Delta x_2, \Delta x_3) = (5 \times 10^{-1}, 5 \times 10^{-1}, 5 \times 10^{-1})$. We compute the field evolution dynamics in this Cartesian coordinate system using the timestep $\Delta t = 10^{-2}$ for $t \in [0, 5]$.

## I.4 A TWO-DIMENSIONAL, INTRINSICALLY CURVED SURFACE

In this case (Section D.3), we work with the FitzHugh-Nagumo system. On the spherical surface, the FitzHugh-Nagumo system written in the $(\theta, \phi)$-coordinate can be described by the following PDEs:

$$
\begin{aligned}
\partial_t V(\theta, \phi, t) = {} & \frac{1}{R^2}\left( \cot\theta\, \partial_\theta + \partial_\theta^2 + \frac{1}{\sin^2\theta}\partial_\phi^2 \right) V(\theta, \phi, t) \\
& + V(\theta, \phi, t) - \frac{1}{3}V^3(\theta, \phi, t) - W(\theta, \phi, t) , \\
\partial_t W(\theta, \phi, t) = {} & \varepsilon\left[ V(\theta, \phi, t) + \beta - \gamma W(\theta, \phi, t) \right] .
\end{aligned}
\tag{49}
$$

We start with the field configuration given by:

$$
\begin{aligned}
V(\rho, \varphi, 0) = {} & V_{01} \exp\left[ -\frac{(\rho - \rho_{V1})^2}{2\sigma_{\rho V1}} - \frac{(\varphi - \varphi_{V1})^2}{2\sigma_{\varphi V1}} \right] \\
& + V_{02} \exp\left[ -\frac{(\rho - \rho_{V2})^2}{2\sigma_{\rho V2}} - \frac{(\varphi - \varphi_{V2})^2}{2\sigma_{\varphi V2}} \right] , \\
W(\rho, \varphi, 0) = {} & W_0 \exp\left[ -\frac{(\rho - \rho_W)^2}{2\sigma_{\rho W}} - \frac{(\varphi - \varphi_W)^2}{2\sigma_{\varphi W}} \right] .
\end{aligned}
\tag{50}
$$

where:

$$
\begin{aligned}
(V_{01}, \rho_{V1}, \varphi_{V1}, \sigma_{\rho V1}, \sigma_{\varphi V1}) &= (1, 140, \pi/6, 15, 0.15) , \\
(V_{02}, \rho_{V2}, \varphi_{V2}, \sigma_{\rho V2}, \sigma_{\varphi V2}) &= (1, 160, \pi/4, 10, 0.1) , \\
(W_0, \rho_W, \varphi_W, \sigma_{\rho W}, \sigma_{\varphi W}) &= (1, 150, \pi/4, 10, 0.2) .
\end{aligned}
\tag{51}
$$

Both $V$ and $W$ are combinations of localized configurations. Then, we numerically integrate the PDEs Eq. (49) for the total amount of time $\Delta T = 2 \times 10^2$ (so that the system has reached the stage of steady spiral wave generation) from the above configurations. We use the final-time configuration as the initial field configuration for the study in Section D.3.

We work on the spherical surface of radius $R = 2 \times 10^2$, analyzing the two-dimensional curved space covered by $(\rho, \varphi) \in [\rho_{\min}, \rho_{\max}] \times [\varphi_{\min}, \varphi_{\max}]$, where $\rho_{\min} = 5 \times 10^1$, $\rho_{\max} = 2 \times 10^2$ and $\varphi_{\min} = 0$, $\varphi_{\max} = \pi/2$. We discretize the coordinates $(\rho, \varphi)$ uniformly with the lattice-spacing

$$(\Delta\rho, \Delta\varphi) = \left( \frac{\rho_{\max} - \rho_{\min}}{300}, \frac{\varphi_{\max} - \varphi_{\min}}{200} \right) . \tag{52}$$

The transformation from these stereographic coordinates $(\rho, \varphi)$ to geographical ones $(\theta, \phi)$ is as described Eq. (28). We use $(\theta, \phi) \in [-L, +L] \times [\theta_{\min}, \theta_{\max}] \times [\phi_{\min}, \phi_{\max}]$ to describe the same space in this new coordinate system, where we use a uniform grid with the lattice-spacing

$$(\Delta\theta, \Delta\phi) = \left( \frac{\theta_{\max} - \theta_{\min}}{300}, \frac{\phi_{\max} - \phi_{\min}}{200} \right) . \tag{53}$$

Note that the $(\rho, \varphi)$-grid and the $(\theta, \phi)$-grids are not comprised of identical sets of physical points. We compute the field evolution dynamics on the spherical surface in this different coordinate system, using the timestep $\Delta t = 10^{-1}$ for $t \in [0, 5 \times 10^1]$.

## I.5 A Two-Dimensional, Flat Surface, with Non-Trivial Topology

We work with the FitzHugh-Nagumo system for this demonstration (Section D.4). In 2-D polar coordinate system, the FitzHugh-Nagumo PDEs can be expressed in the $(\rho, \theta)$-coordinate as:

$$\partial_t V(\rho, \theta, t) = \frac{1}{\rho} \partial_\rho \left[ \rho \partial_\rho V(\rho, \theta, t) \right] + \frac{1}{\rho^2} \partial_\theta^2 V(\rho, \theta, t)$$
$$+ V(\rho, \theta, t) - \frac{1}{3} V^3(\rho, \theta, t) - W(\rho, \theta, t) , \tag{54}$$
$$\partial_t W(\rho, \theta, t) = \varepsilon \left[ V(\theta, \phi, t) + \beta - \gamma W(\theta, \phi, t) \right] .$$

We start with the field configuration given by:

$$V(\rho, \theta, 0) = V_{01} \exp\left[ -\frac{(\rho - \rho_{V1})^2}{2\sigma_{\rho V1}} - \frac{(\theta - \theta_{V1})^2}{2\sigma_{\theta V1}} \right]$$
$$+ V_{02} \exp\left[ -\frac{(\rho - \rho_{V2})^2}{2\sigma_{\rho V2}} - \frac{(\theta - \theta_{V2})^2}{2\sigma_{\theta V2}} \right]$$
$$+ V_{01} \exp\left[ -\frac{(\rho - \rho_{V1})^2}{2\sigma_{\rho V1}} - \frac{(\theta - \theta_{V1} - \pi/2)^2}{2\sigma_{\theta V1}} \right]$$
$$+ V_{02} \exp\left[ -\frac{(\rho - \rho_{V2})^2}{2\sigma_{\rho V2}} - \frac{(\theta - \theta_{V2} - 3\pi/2)^2}{2\sigma_{\theta V2}} \right] , \tag{55}$$
$$W(\rho, \varphi, 0) = W_0 \exp\left[ -\frac{(\rho - \rho_W)^2}{2\sigma_{\rho W}} - \frac{(\theta - \theta_W)^2}{2\sigma_{\theta W}} \right]$$
$$+ W_0 \exp\left[ -\frac{(\rho - \rho_W)^2}{2\sigma_{\rho W}} - \frac{(\theta - \theta_W - \pi/2)^2}{2\sigma_{\theta W}} \right]$$
$$+ W_0 \exp\left[ -\frac{(\rho - \rho_W)^2}{2\sigma_{\rho W}} - \frac{(\theta - \theta_W - 3\pi/2)^2}{2\sigma_{\theta W}} \right] .$$

where:

$$(V_{01}, \rho_{V1}, \theta_{V1}, \sigma_{\rho V1}, \sigma_{\theta V1}) = (1, 140, \pi/6, 15, 0.15) ,$$
$$(V_{02}, \rho_{V2}, \theta_{V2}, \sigma_{\rho V2}, \sigma_{\theta V2}) = (1, 160, \pi/4, 10, 0.1) , \tag{56}$$
$$(W_0, \rho_W, \theta_W, \sigma_{\rho W}, \sigma_{\theta W}) = (1, 150, \pi/4, 10, 0.2) .$$

Both $V$ and $W$ are combinations of localized configurations. Then, we numerically integrate the PDEs Eq. (54) for the total amount of time $\Delta T = 10^3$ (so that the system has reached the stage of

steady spiral wave generation) from the above configurations. We use the final-time configuration as the initial field configuration for the study in Section D.4.

We work on the annulus confined between the radius value $\rho \in [\rho_{\min}, \rho_{\max}]$, where $\rho_{\min} = 10^2$, $\rho_{\max} = 2 \times 10^2$. We discretize the coordinates $(\rho, \theta)$ uniformly with the lattice-spacing

$$(\Delta\rho, \Delta\theta) = (1, 2\pi/800) \ . \tag{57}$$

Then, we compute the field evolution dynamics on the spherical surface with the true dynamics and with the neural network learned dynamics, using the timestep $\Delta t = 10^{-1}$ for $t \in [0, 5 \times 10^1]$.

## J   GEOGRAPHIC AND STEREOGRAPHIC COORDINATES

We consider a sphere of radius $R$, equipped with geographical-coordinates $(\theta, \phi)$ corresponding to spherical-angles, where $\theta = 0$ represents the South pole and $\theta = \pi$ is the North pole. The spherical surface can be embedded in a three-dimensional Euclidean space via Eq. (29), where $(x, y, z)$ are the corresponding Cartesian coordinates. On this curved manifold, the induced metric is given by:

$$ds^2 = R^2 \left( d\theta^2 + \sin^2\theta d\phi^2 \right) \ . \tag{58}$$

There, Laplacians and the inner-products can be calculated from:
- The Laplacians:

$$\star d \star d\Psi_j = \frac{1}{R^2} \left( \cot\theta \partial_\theta \Psi_j + \partial_\theta^2 \Psi_j + \frac{1}{\sin^2\theta} \partial_\phi^2 \Psi_j \right) \ , \tag{59}$$

where $j = 1, 2$. There are two of them.
- The inner products:

$$\langle d\Psi_j, d\Psi_k \rangle = \frac{1}{R^2} \left( \partial_\theta \Psi_j \partial_\theta \Psi_k + \frac{1}{\sin^2\theta} \partial_\phi \Psi_j \partial_\phi \Psi_k \right) \ , \tag{60}$$

where $j, k = 1, 2$. There are three distinct inner products.

We study a region of the sphere surface bounded by latitude $\theta_{\min}$ and $\theta_{\max}$, and by longitude $\phi_{\min}$ and $\phi_{\max}$. We also impose the Neumann (no-flux) conditions as described in Eq. (27) on the latitude boundaries and the periodic conditions on the longitude boundaries.

We focus on a projection that utilizes the South pole as the projection viewpoint and the equator plane as the projection plane. A convenient mapping — written in stereographic polar-coordinates — is as introduced in Eq. (28). Thus, the metric in Eq. (58) after this coordinate transformation becomes:

$$ds^2 = \left[ \frac{4R^4}{(R^2 + \rho^2)^2} \right] \left( d\rho^2 + \rho^2 d\varphi^2 \right) \ . \tag{61}$$

The Laplacians and the inner-products in these new coordinates can be expressed with:
- The Laplacians:

$$\star d \star d\Psi_j = \left[ \frac{(R^2 + \rho^2)^2}{4R^4} \right] \left( \frac{1}{\rho} \partial_\rho \Psi_j + \partial_\rho^2 \Psi_j + \frac{1}{\rho^2} \partial_\varphi^2 \Psi_j \right) \ , \tag{62}$$

where $j = 1, 2$. There are two of them.
- The inner products:

$$\langle d\Psi_j, d\Psi_k \rangle = \left[ \frac{(R^2 + \rho^2)^2}{4R^4} \right] \left( \partial_\rho \Psi_j \partial_\rho \Psi_k + \frac{1}{\rho^2} \partial_\rho \Psi_j \partial_\rho \Psi_k \right) \ , \tag{63}$$

where $j, k = 1, 2$. There are three distinct inner products.

In this coordinate system, the spatial domain of interests is bound between radius $\rho_{\min}$ and $\rho_{\max}$, and angle $\varphi_{\min}$ and $\varphi_{\max}$, which can be determined from $\theta_{\min}, \theta_{\max}, \phi_{\min}, \phi_{\max}$, with Eq. (28). Figure 6A-C illustrates how geographical coordinates $(\theta, \phi)$ and stereographic coordinates $(\rho, \varphi)$ are related through a projection from the South pole onto the equatorial plane. It shows that the same region on the surface of the sphere can be covered by evenly-spacing grids in both coordinate systems, even though these grids are not equivalent.

We aim to demonstrate that, using the neural network model trained in one dimension in Sec. E, we can predict the correct field evolution dynamics on the surface of the sphere regardless of the chosen coordinate system. In other words, not only do we need to generate a prediction, but also compare it with the expected true dynamics. We begin with an initial condition defined in $(\theta, \phi)$-space with uniform $(\Delta\theta, \Delta\phi)$ discretization. We then compute the field evolution in the $(\theta, \phi)$-coordinate using direct integration of the true PDE. Next, we determine by extrapolation the corresponding initial condition in $(\rho, \varphi)$-space with uniform $(\Delta\rho, \Delta\varphi)$ discretization and use it to generate the field evolution in $(\rho, \varphi)$-coordinates via the trained neural network model in Section E. To make comparison, we transform both the generated datasets to the three-dimensional Euclidean $(x, y, z)$-space. These steps are similar to what has been demonstrated for one-dimensional case, see Fig. 9. For our spatially liberated neural network model, all these generated datasets are expected to be in agreement, which we confirm through our results.

## K  COMPARING "TRUE" AND LEARNED DYNAMICS

To compare the "true" dynamics $\Psi_i(\zeta, t)$ with the neural learned dynamics $\hat{\Psi}(\eta, t)$ (which computed at different locations), we extrapolate from $\hat{\Psi}$ evaluated at $\eta$-locations to its corresponding values at $\zeta$-locations, denoted as $\tilde{\Psi}_i(\eta, t)$. We then can calculate $\Psi_i - \tilde{\Psi}_i$ to see the deviation between these dynamics and report the mean square-error (MSE) for all cases.

### K.1  1-, 2- AND 3-DIMENSIONAL EUCLIDEAN SPACE

We show the deviation for Patlak-Keller-Segel system in one-dimensional Euclidean space (Fig. 10A), FitzHugh-Nagumo system in two-dimensional Euclidean space (Fig. 10B), and Barkley system in three-dimensional Euclidean space (Fig. 10C).

We report the MSE for each case shown in Fig. 10, using $\text{MSE}_{\Psi_i}$ to denote the MSE of the field $\Psi_i$ that is compared between the "true" and "spatially liberated" neural-learned dynamics:
- Fig. 10A1: $\text{MSE}_A = 1.67 \times 10^{-5}$, $\text{MSE}_B = 4.90 \times 10^{-6}$.
- Fig. 10A2: $\text{MSE}_A = 8.56 \times 10^{-6}$, $\text{MSE}_B = 7.76 \times 10^{-5}$.
- Fig. 10A3: $\text{MSE}_A = 1.24 \times 10^{-5}$, $\text{MSE}_B = 7.98 \times 10^{-5}$.
- Fig. 10B1: $\text{MSE}_V = 2.01 \times 10^{-3}$, $\text{MSE}_W = 1.20 \times 10^{-5}$.
- Fig. 10B2: $\text{MSE}_V = 1.62 \times 10^{-2}$, $\text{MSE}_W = 1.49 \times 10^{-4}$.
- Fig. 10B3: $\text{MSE}_V = 3.03 \times 10^{-2}$, $\text{MSE}_W = 9.16 \times 10^{-4}$.
- Fig. 10B4: $\text{MSE}_V = 4.67 \times 10^{-2}$, $\text{MSE}_W = 1.39 \times 10^{-3}$.
- Fig. 10C1: $\text{MSE}_U = 0$, $\text{MSE}_V = 0$ (initial field configurations are identical).
- Fig. 10C2: $\text{MSE}_U = 3.19 \times 10^{-3}$, $\text{MSE}_V = 3.35 \times 10^{-4}$.
- Fig. 10C3: $\text{MSE}_U = 6.24 \times 10^{-3}$, $\text{MSE}_V = 7.44 \times 10^{-4}$.
- Fig. 10C4: $\text{MSE}_U = 9.31 \times 10^{-3}$, $\text{MSE}_V = 1.15 \times 10^{-3}$.

### K.2  2-DIMENSIONAL INTRINSICALLY CURVED SURFACE

We show the deviation for FitzHugh-Nagumo system in two-dimensional curved space, which is embedded in three-dimensional Euclidean space, in Fig. 11.

We report the MSE for each time frame shown in Fig. 11, using $\text{MSE}_{\Psi_i}$ to denote the MSE of the field $\Psi_i$ that is compared between the "true" and "spatially liberated" neural-learned dynamics:
- Fig. 11A: $\text{MSE}_V = 5.85 \times 10^{-5}$, $\text{MSE}_W = 1.35 \times 10^{-8}$.
- Fig. 11B: $\text{MSE}_V = 1.36 \times 10^{-3}$, $\text{MSE}_W = 1.51 \times 10^{-5}$.
- Fig. 11C: $\text{MSE}_V = 2.34 \times 10^{-3}$, $\text{MSE}_W = 6.56 \times 10^{-5}$.
- Fig. 11D: $\text{MSE}_V = 4.47 \times 10^{-3}$, $\text{MSE}_W = 1.01 \times 10^{-4}$.

### K.3  2-DIMENSIONAL NON-TRIVIAL TOPOLOGY

We show the deviation for FitzHugh-Nagumo system in two-dimensional flat space with an annulus topology, in Fig. 12.

We report the MSE for each time frame shown in Fig. 11, using $\text{MSE}_{\Psi_i}$ to denote the MSE of the field $\Psi_i$ that is compared between the "true" and "spatially liberated" neural-learned dynamics:

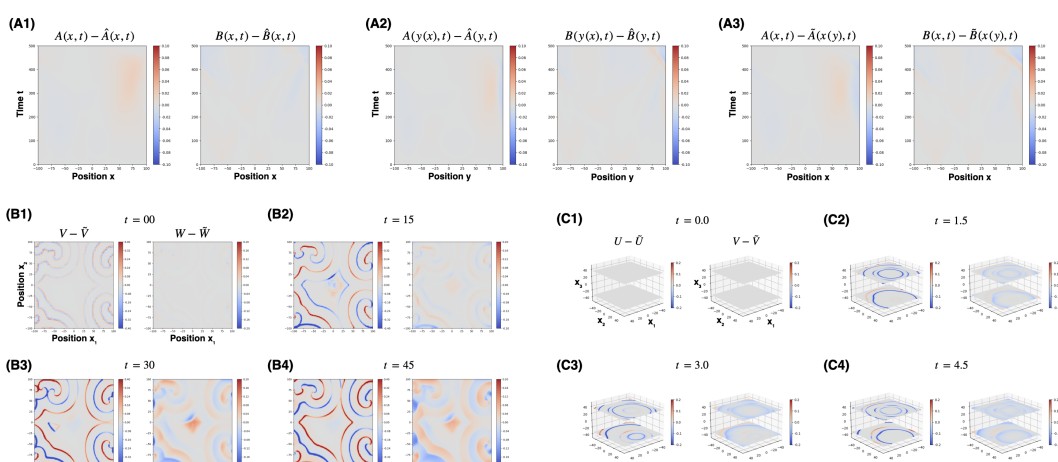

**Figure 10: Comparisons between "true" and neural learned dynamics in Euclidean spaces.** The deviation range for each field is equal to 10% its dynamical range, with negative deviation colored red-ish and positive deviation colored blue-ish. For Patlak-Keller-Segel system, we show the deviation between Fig. 2B1 and Fig. 2B2 in **(A1)**, between Fig. 2C and Fig. 2D1 in **(A2)**, and between Fig. 2B1 and Fig. 2D2 in **(A3)**. For FitzHugh-Nagumo system, we show the deviation between the top and bottom-rows in Fig. 3A1-4 in **(B1-4)**, respectively. For FitzHugh-Nagumo system, we show the deviation between the top and bottom-rows in Fig. 3B1-4 in **(C1-4)**, respectively. Note that, for brevity, we only label the first plot in **(B)** and **(C)**, since the rest are identical.

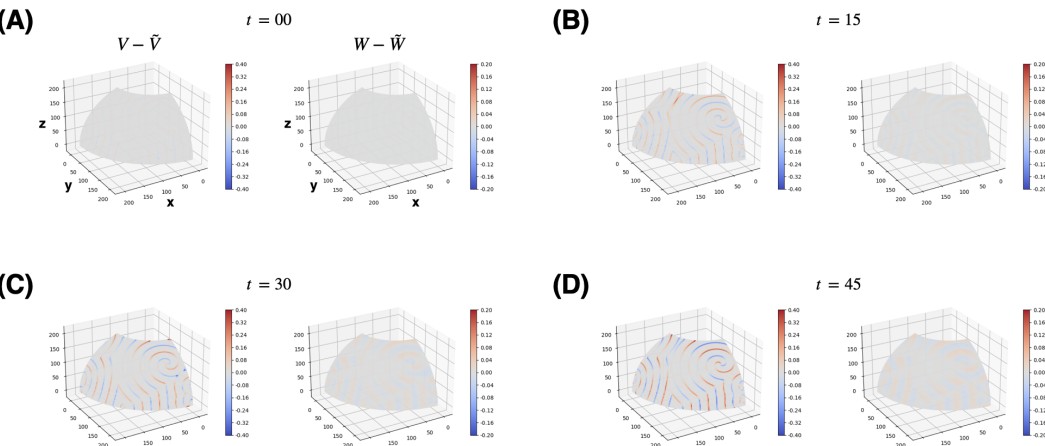

**Figure 11: Comparisons between "true" and neural learned dynamics in two-dimensional intrinsically curved surface embedded in three-dimensional Euclidean space.** Here we study FitzHugh-Nagumo system, where the deviation range for each field is equal to 10% of its dynamical range, with negative deviation colored red-ish and positive deviation colored blue-ish. We show the deviation between the top and bottom rows of Fig. 6D1 in **(A)**, of Fig. 6D2 in **(B)**, of Fig. 6D3 in **(C)**, and of Fig. 6D4 in **(D)**. Note that, for brevity, we only label the first plot, since the rest are identical.

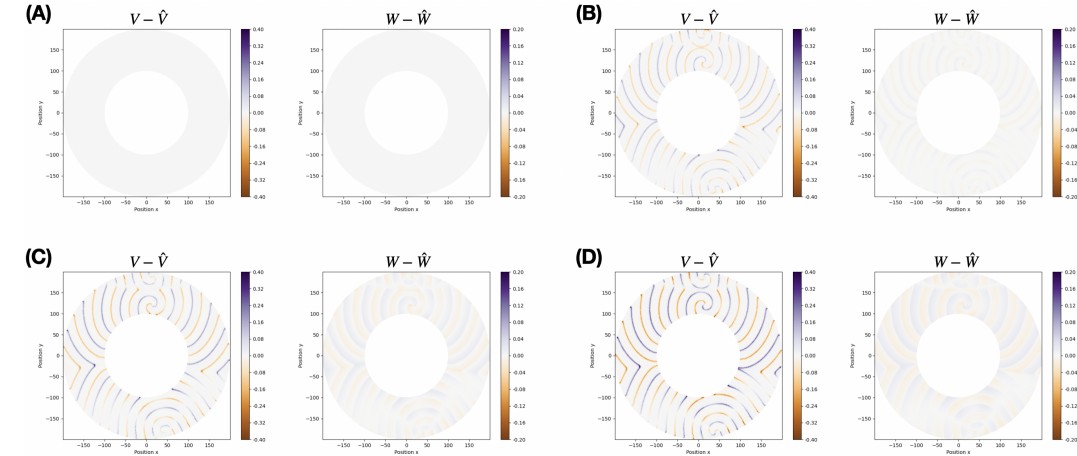

**Figure 12: Comparisons between "true" and neural learned dynamics in two-dimensional flat surface embedded with an annulus topology.** Here we study FitzHugh-Nagumo system, where the deviation range for each field is equal to 10% of its dynamical range, with negative deviation colored red-ish and positive deviation colored blue-ish. We show the deviation between the top and bottom rows of Fig. 7A in **(A)**, of Fig. 7B in **(B)**, of Fig. 7C in **(C)**, and of Fig. 7D in **(D)**.

- Fig. 12A: $\text{MSE}_V = 0$, $\text{MSE}_W = 0$ (initial field configurations are identical).
- Fig. 12B: $\text{MSE}_V = 9.25 \times 10^{-4}$, $\text{MSE}_W = 1.11 \times 10^{-5}$.
- Fig. 12C: $\text{MSE}_V = 1.64 \times 10^{-3}$, $\text{MSE}_W = 4.69 \times 10^{-5}$.
- Fig. 12D: $\text{MSE}_V = 3.33 \times 10^{-3}$, $\text{MSE}_W = 7.54 \times 10^{-5}$.

## L ON THE QUALITY OF THE PREDICTION USING NOISY TRAINING DATA

In fact, our study was developed with real experiments in mind: we are preparing to apply it to a real bacteria-chemical system as reported in Phan et al. (2024), which is also the work that proposed the modified Patlak-Keller-Segel PDE used in the main text. The model in Eq. (24) is presented in dimensionless form; Table 1 summarizes the mapping between dimensional parameters and their dimensionless counterparts.

| Asp dynamics | Bacteria dynamics | Non-Dimensionalized |
|---|---|---|
| $D_A = 800\mu\text{m}^2/\text{s}$ | $D_B \approx 440\mu\text{m}^2/\text{s}$ | $D_B \approx 0.55$ |
| $\gamma_0 \approx 0.53\mu\text{M/OD/s}$ | $\chi_0 \approx 3800\mu\text{m}^2/\text{s}$ | $\chi_0 \approx 4.8$ |
| $B_C \approx 5.1\text{OD}$ | $B_h \approx 6.8\text{OD}$ | $B_h \approx 1.3$ |
| $A_h \approx 3.7\mu\text{M}$ | $K_i \approx 1\mu\text{M}$ | $K_i \approx 0.27$ |
| | $\alpha \approx 0.4/\text{h}$ | $\alpha \approx 1.5 \times 10^{-4}$ |

**Table 1:** The best-estimated parameters are those reported in Phan et al. (2024), based on an analysis of traveling-wave experiments with *E. coli* strain HE205 Cremer et al. (2019). Unlisted non-dimensionalized parameters are, by convention, implied to be set to 1, i.e. $D_A = 1$, $\gamma_0 = 1$, $B_C = 1$, and $A_h = 1$. In those experiments, the maximum Asp concentration $\approx 100\mu\text{M}$ and the maximum cell density $\approx 6\text{OD}$, which correspond to the nondimensional maxima $A_{\max} \approx 27$ and $B_{\max} \approx 1$. The non-dimensionalized unit-time and unit-length correspond to the time-scale $\tau \approx 1.4\text{s}$ and the length-scale $\lambda \approx 33\mu\text{m}$. While these values can vary across strains, they are expected to remain within the same order of magnitude.

We can show that the experimental noise for this system is relatively low, can be made much lower, and also can be estimated as follows. The microfluidic channel has width $W \approx 1.2\text{mm}$ and depth $H \approx 100\mu\text{m}$ (which can be increased). The bacterial density in the traveling wave is around $n \sim 2\text{OD}$ (with $1\text{OD} \approx 8 \times 10^8\text{cells/mL}$), and wave peaks can reach $\sim 5\text{OD}$. The effective one-dimensional bin size in our discretization is $\Delta x = 1$ (dimensionless), corresponding to $\approx 33\mu\text{m}$ physically (see the caption of Table 1). Hence, the expected number of bacteria per bin is

$$N \sim nWH\Delta x \approx 8 \times 10^3 \text{cells} .$$

Assuming purely Poisson (counting) statistics, which approach a Gaussian in the large-number limit, the relative shot noise in the bacterial density field $B(x,t)$ is $\sigma \sim N^{-1/2} \sim 1\%$. Using a deeper device, e.g. increasing $H$ to $1.2\text{mm}$, would reduce this further to $\sigma \approx 0.3\%$. The chemical concentration field $A(x,t)$ is even less noisy (effectively negligible in experiments), owing to its much higher molecular counts and diffusivity compared to those of bacteria. We believe these empirical conditions make the noise-free assumption acceptable here, while still laying groundwork for further studies.

If we add this $\sigma \sim 0.3\%$ noise to the training data and apply Savitzky-Golay smoothing before computing derivatives, the neural network – after the same training mentioned in Appendix E.3 – can still make reasonable and correct-qualitative predictions (especially in the region where both bacteria and chemical field values are high, where noise does not dominate training, which is from $t = 0$ to 250) for the initial condition as used in Fig. 2B1 and Fig. 2B2 of the main manuscript. Here we report the MSE for the values between $t = 0$ and $t = 250$ (see Fig. 13A):

- $\text{MSE}_A = 4.76 \times 10^{-3}$, $\text{MSE}_B = 1.24 \times 10^{-3}$.

Despite a substantially higher MSE after adding noise, the practical implication does not change: the prediction correctly tracks the traveling-wave front (speed, shape, persistence). Also, it should be noted that our current noise-handling strategy is simple – simply applying a mild smoothing filter before training – but already yields acceptable predictions.

There are several ways to improve performance; for instance, changing the training procedure by switching from an $L^2$-norm to a $L^1$-norm for the loss function makes the neural network model much more robust to noise. For comparison, we report this MSE between $t = 0$ and $t = 250$:

- $\text{MSE}_A = 5.26 \times 10^{-4}$, $\text{MSE}_B = 1.32 \times 10^{-4}$.

These results are now smaller by more than an order of magnitude and could be reduced further with more sophisticated noise-resistant strategies.

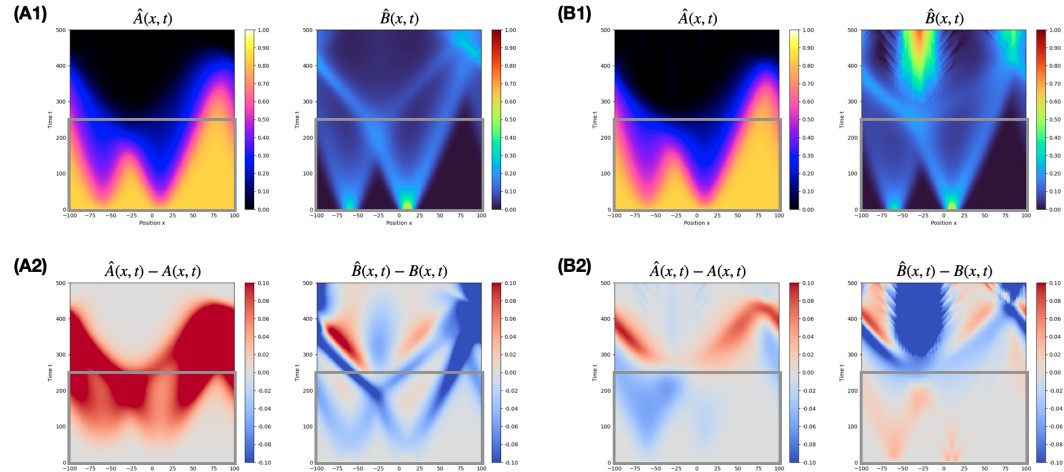

**Figure 13: The predictions for Patlak-Keller-Segel field evolution dynamics in a periodic one-dimensional space, using training data corrupted by realistic noise.** The initial field configurations are the same as used in Fig. 2B1 and Fig. 2B2. The gray boxes capture the spacetime regions between $t = 0$ and $t = 200$. **(A1)** The time progression of the fields when using the neural network trained the same way as in Appendix E.3. **(A2)** The field values difference between the neural network predicted Fig. 13A1 and the true Fig. 2B1. **(B1)** The time progression of the fields when using the neural network trained with $L^1$-norm (instead of $L^2$-norm). **(B2)** The field values difference between the neural network predicted Fig. 13B1 and the true Fig. 2B1.



**Figure 14: Left:** Solution of the Helmholtz equation obtained using classical PINN training. **Middle:** Solution of the Helmholtz equation obtained using a PINN with a coordinate-free PDE loss, where the PDE operator is identified from trajectories of the 1-D harmonic oscillator (see text for details). **Right:** Squared error between classical PINN solution and coordinate-free PINN solution to the 2D helmholtz problem.

## M GENERAL PDE EXAMPLE

### M.1 COMPUTATIONAL DETAILS

The PCA model was fit to the 200 numerical trajectories of the 1-D harmonic oscillator. For these solutions, we approximate the first and second derivatives of $u$ using finite differences on a set of 200 solutions (with random initial conditions), and perform PCA, we obtained explained variance ratios of $6.7e - 1, 3.3e - 1, 6.4e - 6$, denoting the 2-dimensional linear structure of the manifold.

After identifying the operator, we proceed to train a PINN in the manner discussed in the main text. The PINN architecture consists of three layers with width 64 and a `tanh` activation function, trained using Adam Kingma (2014). The mean square error between a classical PINN solution and the coordinate-free version visualized in Fig. 14 is approximately $4e^{-4}$, while the square difference between the these solutions is depicted the right sub-plot of Figure 14.

### M.2 TRANSFERABILITY OF BOUNDARY CONDITIONS

Boundary conditions can be also be seen as algebraic relationships between differential terms, *now on the boundary of a manifold $\mathcal{M}$*. In particular, given an embedded Riemannian manifold $\mathcal{M}$, then

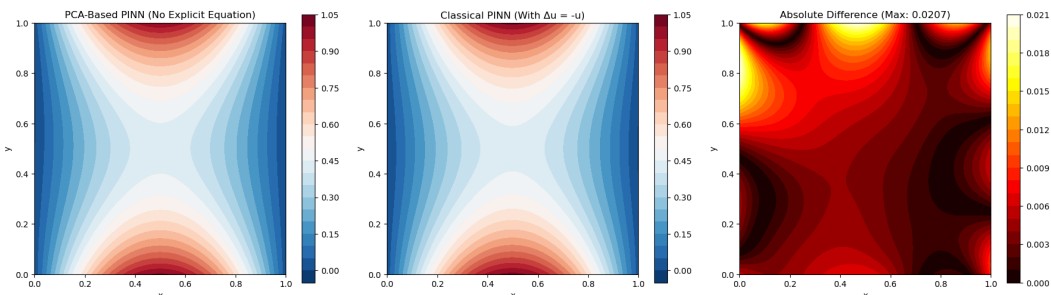

**Figure 15:** Comparison between **(left)** transferred PDE+Boundary conditions for the Helmholtz Equation, and **middle** classical PINN with known boundary conditions. **Right** Absolute error between the known-PINN and transferred-PINN solutions

its boundary $\partial \mathcal{M}$ is also a Riemannian manifold of one dimension less, thus having a well-defined normal vector.

For a PDE written on $\mathcal{M}$, the boundary condition can be seen as an algebraic relationship between differentials of the fields involved in the PDE and an externally specified set of boundary fields. Intrinsic boundary fields include:

$$B_{\text{boundary}} = \left\{ u_i|_{\partial \mathcal{M}}, \langle \mathbf{n}, \boldsymbol{\nabla} u_i|_{\partial \mathcal{M}} \rangle, \Delta u_i|_{\partial \mathcal{M}}, f_i^{bc} \right\}_{i=1}^N \tag{64}$$

where $\mathbf{n}$ is the normal vector (pointing either to the interior or exterior of $\mathcal{M}$ by convention), and $f_i^{bc}$ is a user-specified set of boundary fields.

For example, for a simple Dirichlet boundary condition for a scalar field $u$, we would have $f^{bc} = 0$, and the boundary condition relationship between these fields is linear: $u|_{\partial \mathcal{M}} = f^{bc}$. This type of boundary condition is, therefore, transferable, and can be encoded agnostically in the same way as the PDE law itself.

**Example: Helmholtz with transferred boundary condition:** We repeat the Helmholtz example in the case where the boundary condition is not explicitly known. We construct two coordinate-free libraries, $B_{\text{interior}}$ which is the PDE feature basis, and $B_{\text{boundary}}$ as described in this section. We populate these libraries using finite-difference features from the 1-D Helmholtz harmonic oscillator, and perform PCA on the resulting point clouds to encode the (in this case linear) relationships between the basis elements. We proceed to train a PINN, minimizing the corresponding two-term loss. Both losses are computed using the projection approach of 3.2. The relative $L^2$ error between the transferred PINN solution (including boundary conditions) and the PINN solution of the known equation and boundary conditions in 2D is approximately $1.3\%$. The solutions demonstrated in 15 are in good agreement.

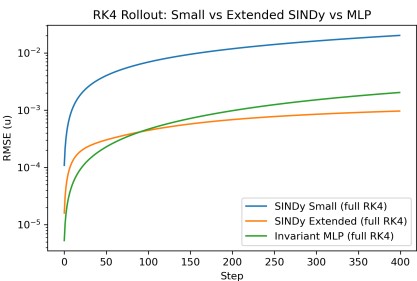 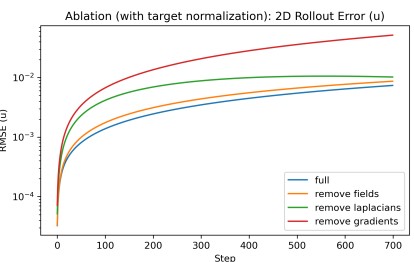

**Figure 16: (Left)** Roll-out errors for SINDy and MLP identification methods using RK4. **Right:** Roll-out error for MLP identification, using ablated feature libraries.

## N    VARIABLE-COEFFICIENT DIFFUSION

The coordinate-free formulation can handle PDEs with time-variable coefficients, by treating them as additional scalar fields that participate in the construction of the basis.

### N.1    EXAMPLE

We consider a non-linear diffusion equation with a variable diffusion coefficient:

$$\frac{\partial u}{\partial t} = \boldsymbol{\nabla}(f(x,t)\boldsymbol{\nabla}u) + u(1-u) \tag{65}$$

$$\frac{\partial f}{\partial t} = \alpha\Delta f + \beta u^2 \tag{66}$$

with $\alpha = 0.01$, $\beta = 0.1$. $u$ and $f$ are treated as separate scalar fields. Note that, in other variations, one may remove the coupling term $\beta u^2$ from the second equation, so that the coefficient evolution is independent, or simply specify $f(x)$ without time dependence, for a purely spatially-varying coefficient.

We collect data in 1-D, integrating the corresponding ODE with 40 separate realizations with randomly chosen initial conditions, after which we use finite differences to estimate the basis $B$. We then use both an MLP model (predicting $\frac{\partial u}{\partial t}, \frac{\partial f}{\partial t}$ from $B$) and two symbolic (SINDy) models to extract the relationship between the basis variables. Of the two SINDy libraries, one is smaller ($L_{\text{small}}$) than the other ($L_{\text{extended}}$), showcasing the different behavior of the symbolic method, which is sensitive to the size of the library and the way in which the nonlinearity is captured.

$$L_{\text{small}} = \left\{ 1, u, u^2, u^3, f, (\frac{\partial u}{\partial x})^2, \frac{\partial^2 u}{\partial x^2} \right\}$$

$$L_{\text{extended}} = \left\{ 1, u, u^2, u^3, f, \|\boldsymbol{\nabla}u\|^2, \Delta u, f\Delta u, \langle\boldsymbol{\nabla}f, \boldsymbol{\nabla}u\rangle \right\}$$

We now integrate a 2-D system with a Gaussian initial field $u$ and randomly generated initial coefficient $f$ using an RK4 integrator with each of the three models. The roll-out integration error is presented in figure 16

Because the interaction between $u$ and $f$ is nonlinear, one must construct an extended library to capture the true relationship between the variables; given the right library, extracting a correct symbolic relation is successful. However, if the relationship is inherently complicated, the size of the required library increases dramatically. On the other hand, the MLP is nonlinear, and will therefore not generalize outside the training domain; yet, its ability to capture non-linear relationships between the basis elements enables us to only use a compact library that generates the dynamics. There are tradeoffs in interpretability, transferability, and complexity, when using either method.

In addition, we perform an ablation study for each type of feature (fields, gradient inner products, Laplacians) in the case of the MLP network. We compare training a full-library MLP with three libraries where each type of feature is removed (and where the features are normalized). The numerical roll-out and final errors are presented in figure 16. We observe that while all features

contribute to the preddiction throught the integration window, the gradient and Laplacian features play a more significant role, especially in earlier time steps.

## O    HIGHER-ORDER LIBRARIES

In the main text, we explore the coordinate- and dimension-free learning framework using libraries constructed with up-to-second order terms. Libraries with arbitrary higher-order terms can subsequently be constructed by recursively combining lower-order library terms, via the gradient inner product and Laplacian operators.

Let $k$ denote the order of the library $B_k$. For a collection of scalar fields $\{u_i\}_{i=1,N} \equiv B_0$, we define $B_2$ as the basis appearing in the main text, consisting of terms of the form

$$B_2 = \{u_i, \langle \boldsymbol{\nabla} u_i, \boldsymbol{\nabla} u_j \rangle, \Delta u_i\}_{i=1}^N$$

. To construct a library of *any* order, we then define the recursive relation:

$$B_{k+1} = \{v_i, \langle \boldsymbol{\nabla} v_i, \boldsymbol{\nabla} v_j \rangle, \Delta v_i\}_{i=1}^N, \quad v_i \in B_k. \tag{67}$$

The size $s_k$ of the library of order $k$ will scale polynomially in $k$. We note that, while the construction of the library is analytically tractable, numerical integration of higher-order systems becomes generically harder as the dimension increases.

**Example: 4th order library for a single scalar field:** Let $u_i : \mathbb{R}^n \to \mathbb{R}$ be a scalar field. Then:

$B_0 = \{u\}$

$B_2 = \left\{u, \|\boldsymbol{\nabla} u\|^2, \Delta u\right\}$

$B_4 = \left\{u, \|\boldsymbol{\nabla} u\|^2, \Delta u, \left\langle \boldsymbol{\nabla} u, \boldsymbol{\nabla} \|\boldsymbol{\nabla} u\|^2 \right\rangle, \langle \boldsymbol{\nabla} u, \boldsymbol{\nabla} \Delta u \rangle, \left\langle \boldsymbol{\nabla} \|\boldsymbol{\nabla} u\|^2, \boldsymbol{\nabla} \Delta u \right\rangle, \Delta \left(\|\boldsymbol{\nabla} u\|^2\right), \Delta^2 u\right\}.$

**Example: The Cahn-Hilliard Equation** We apply our methodology to the 4-th order Cahn-Hilliard Equation:

$$\frac{\partial u}{\partial t} = \alpha \Delta (u^3 - u - \beta^2 \Delta u) \tag{68}$$

with $\alpha, \beta \in \mathbb{R}$. In our transfer framework, we solve the 1-D system for $u(x,t)$ on the interval $x \in [0,6]$, with $\alpha = 1, \beta = 0.1$, using a spectral solver. We compute the higher-order coordinate-free library (using the spectral representation of the solution), and train a fully connected neural network to predict $u_t$ in terms of the features. We then use an Euler solver in a 2-D square, to integrate for short time-steps, showing transferability of the learned right-hand-side. The relative $L^2$ error remains around 10% throughout this integration. Importantly, the transferred dynamics demonstrate the (seemingly) correct qualitative structure.

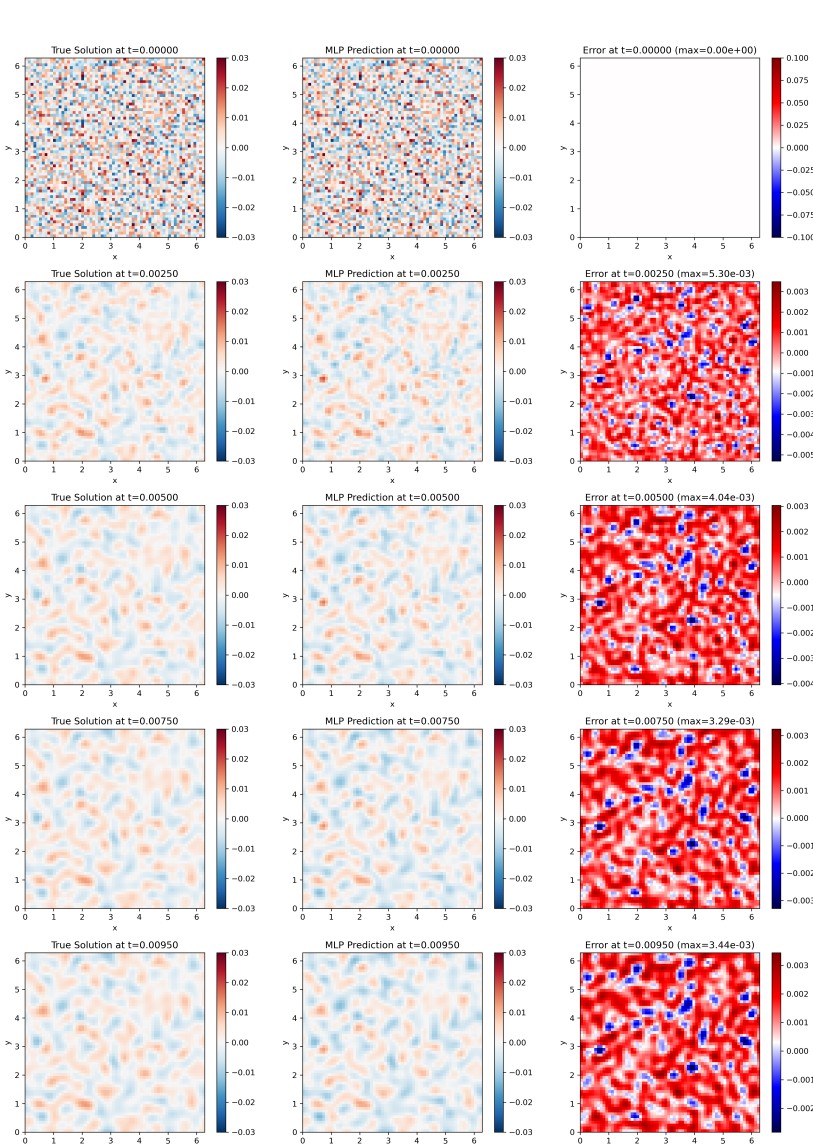

**Figure 17:** Euler roll-out for the Cahn-Hilliard Equation, transferred from 1-D to a 2-D square with randomly sampled initial condition.

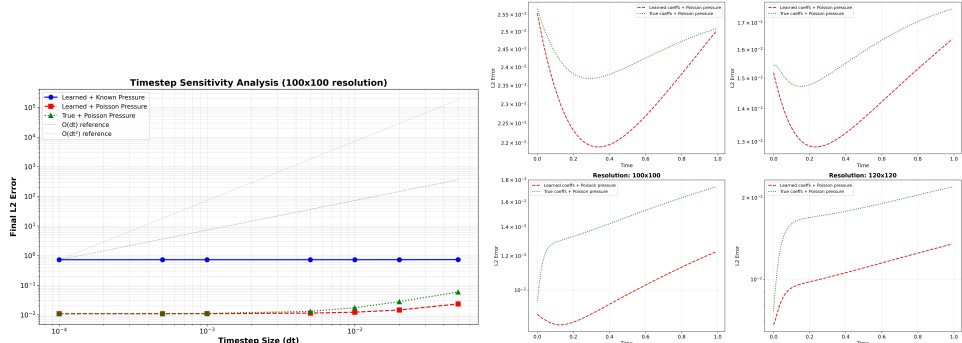

**Figure 18:** Rollout analysis for the Taylor-Green transfer problem. **(Left)** $L^2$ error at time $t = 1.0$ between the true and numerically integrated trajectories using an Euler integrator for varying timesteps $\Delta t$. **(Right)** Rollout error comparison between true and learned coefficient integration at various grid resolutions for $\Delta t = 0.01$.

## P   VECTOR PDE EXAMPLE

The solution to the flow around a cylinder example is computed in a square domain, $(x, y) \in [0, 1]^2 \subset \mathbb{R}^2$, discretized using a regular grid of $100 \times 100$ points. A circle (cylinder in 3D) of radius $r = 0.1$ is centered at the center of the domain. The velocity field $v(x, y, t)$ is set initially to zero $v(x, y, 0) = (0, 0)$, with the exception of the left boundary of the square where we set $v(0, x, t) = 1$. The viscosity $\nu = \eta / \rho$ is set to $\nu = 0.01$.

After obtaining a numerical solution $\hat{v}$ using the setup described above, we compute the corresponding vector features $\{(v \cdot \boldsymbol{\nabla})v, \Delta v, \boldsymbol{\nabla}p\}$. Using least squares, we determine the coefficients that best approximate $\frac{\partial v}{\partial t}$ as a linear combination of these features, which for the example presented in Figure 5 are $(1.01, 0.01, 1.00)$ (rounded to decimal digits).

Using these coefficients, we integrate the momentum equation for the Taylor-Green vortex problem described below, using an Euler integration scheme with a Poisson projection step to handle the unknown pressure field. We integrate on the domain $[0, 2\pi]^2 \subset \mathbb{R}^2$, discretized over various grid sizes $(50 \times 50, 70 \times 70, 100 \times 100, 120 \times 120)$. Overall, we observe that the roll-out error given the learned coefficients is comparable to integrating the equation using the *true* coefficients when the Euler step is small $(\Delta t < 10^{-2})$, while the learned coefficients perform better when the step is larger $(\Delta t > 10^{-2})$; this is a consequence of the learned coefficients accounting for the integrator bias when discretization is coarse, and is an expected phenomenon, whose precise nature is specific to the PDE example and integrator used.

**The Taylor-Green Vortex dynamics** (Taylor & Green, 1937) are a special case of the incompressible Navier-Stokes equation, defined as follows:

For the (periodic) domain $(x, y) \in [0, 2\pi]^2$, we consider the initial velocity field to take the form

$$v_1 = \cos(x)\sin(y), \quad v_2 = -\sin(x)\cos(y) \tag{69}$$

In this case, it is *known* that the velocity field as a function of time takes the form:

$$v(x, y, t) = v(x, y, 0)e^{-2\frac{\eta}{\rho}t} \tag{70}$$

$$p(x, y, t) = \frac{\rho}{4}(\cos(2x) + \cos(2y))e^{-4\frac{\eta}{\rho}t} \tag{71}$$

To further simplify the PDE identification and integration problem, we assume that the scalar field $p(x, y, t)$ is *known* at all times, and that the continuity equation is also known and does not need to be identified. In that way, we have reduced the problem to identifying a time-evolution vector-PDE (i.e. the momentum equation) for which we can use a simple Euler scheme to integrate (in order to verify our coordinate-free learning procedure).

