# OpenReview forum: "Towards Coordinate- and Dimension-Agnostic Machine Learning for Partial Differential Equations"
_ICLR.cc/2026/Conference — Submitted to ICLR 2026_

### Official Review · Reviewer_tW7v · 2025-10-26

**Soundness:** 3
**Presentation:** 2
**Contribution:** 3
**Rating:** 6
**Confidence:** 4

**Summary:**

This paper introduces a methodology for data-driven discovery of PDE models that is agnostic to the spatial domain, coordinate system, spatial dimension, and more. The approach is based on computing analytically known local features from full field observation data. Neural networks are then trained to map these features to the right hand side dynamics of the PDE. A suite of numerical experiments in 1 to 3 dimensions and various geometries are presented that support the potential of the approach.

**Strengths:**

The paper is very original. It nicely combines ideas from different areas of mathematics to produce an implementable numerical framework. The idea of spatially liberated scientific machine learning is highly significant, even if the PDEs are known because it could accelerate training data generation (train in 1D, test in higher dimensions). The writing is of adequate quality and mostly clear.
The authors' methodology is well thought out and some limitations are addressed. For example, I like how in lines 203--208 the well-posedness of the PDEs (e.g., in terms of the required boundary conditions for Helmholtz) are mentioned when discussing generalization to higher dimensions. However, I find the notation hard to parse at times, and the authors do not emphasize the locality of the neural networks and the data.

**Weaknesses:**

While the paper does succeed in showing that 1D training data can provide information about solutions in higher dimensions (under appropriate assumptions on the PDEs), beyond this the numerical results as presented in the main text are not super convincing.
The results are all qualitative and show ``eyeball norm'' only. No actual function space norms of the error are computed or reported until the appendix (the vector PDE Navier-Stokes example has a large 12\% relative error, which is not reported in the main text). Even then, it would be informative to understand what is lost by transferring 1D solutions to higher dimension or coordinates. For example: how much does the relative error change? At what rate? The paper would be strengthened if error tables or convergence rate plots are presented, instead of showing eyeball norm images of fixed problem instances. If the method is not very accurate, could it still be useful? Say, as a preconditioner? Or could one dynamically correct the learned model with data assimilation?

There is not much discussion of the limitations of the proposed framework. For instance, due to the locality required, the approach will not be able to handle integro-differential equations such as the kinetic Boltzmann equation (which has a nonlocal collision operator). One way to get around this is to use a neural operator instead of a fully connected NN in item 2 line 299 page 6. This would allow going from pointwise local maps (Nemytskii operators) to fully nonlocal operators. It is also important for the authors to describe the failure modes of the approach. For example, is it sensitive to model error in the dynamics?

**Questions:**

Questions:
1. The invariant features in Eqn (1) are selected from prior knowledge and ``hard-coded''. Would there be scenarios where $ B $ should itself be learned from the training data?
1. Although any-dimensional machine learning ideas motivate the work, I did not see mention of any-dimensional architectures being used by the authors (e.g., DeepSets, PointNet). Is there an opportunity to blend what's done in the paper and these methods?
1. The particular form of the data is not discussed in detail; it is just vaguely referred to as field data. Although the appendix considers noisy data, what about sparse measurements in space and time instead of a full field grid?
1. The amount of data is also not discussed often. Even in Appendix N, the amount of training data generated is not reported. Relating to Weaknesses, it would be very interesting to understand the data requirements of the approach. For example, consider training on $ N $ 1D data fields versus training directly on $ M $ 2D data fields/trajectories. Always compute test error in 2D. How is $ N $ related to $ M $? This is a very interesting sample complexity question.

Typos:
1. In Eqn 3 page 5, what is $ v_i $? And is $ f_iv_i $ supposed to be interpreted as pointwise multiplication? Eqn 3 is very hard to understand.
1. line 303 p 6 typo: spatial not spacial
1. Typo line 696 p. 13 $ \Delta x $ not $ dx $ in numerator
1. In line 230, why is $ \langle v, \nabla\cdot v\rangle  $ vector-valued?

---

### Official Review · Reviewer_YLLF · 2025-10-28

**Soundness:** 2
**Presentation:** 1
**Contribution:** 4
**Rating:** 4
**Confidence:** 2

**Summary:**

The authors reformulate the problem in terms of coordinate- and dimension-independent representations, paving the way to spatially liberated PDE learning. This allows us to learn a differential equation in low-dimensional spaces and generalize to higher-dimensional spaces with different geometric properties. The proposed approach is summarized in Figure 1. Numerical experiments demonstrate that the proposed approach allows for seamless transitions across various geometric contexts.

**Strengths:**

* (Technical contribution) The paper shows that the dynamics learned in one space can be used to make accurate predictions in other spaces with different dimensions, coordinate systems, boundary conditions, and curvatures.
* (Technical contribution & Novelty) The paper push previous ideas further and enables PDE learning not only in a coordinate-independent way but also independently of the data domain, dimension, and geometry: a novel contribution.

**Weaknesses:**

* The proposed approach can be applied only to invariant systems under local orthogonal transformations and diffeomorphic translations.
  * How can we remove this resstriction?
* The paper, Section 2 in particular, is hard to follow, requiring an extensive revision.
  * How can we compute the Laplacians and inner products of the gradients (lines 166-168) from the "given data" (line 150)?
    * I assume the "given data" are physically observable data and are the fields \\Psi_j. Is this correct?
    * Do the authors use symbolic programming?
    * Also, an elaboration on the footnote 3 is appreciated.
  * (Line 188-189) What does it mean by "independent" and "dependent"?
  * (Figure 1) What are the inputs to the neural network? Are they static numbers (scalar or vector values), symbols, or functions of x at t (fields)? If they are functions of x, how can we input functions to a neural network (e.g., is space discretized?)? Or using a symbolic neural network?
  * The explanation in Section 2 is not self-contained, and some contents in Section 3 (e.g., lines 368-373) should be merged.

## Minor comments
* Use the term "PDE" or "partial differential equation" consistently.

**Questions:**

See above.


## Review summary

The motivation and goal are clear, and the contribution (PDE learning not only in a coordinate-independent way but also independently of the data domain, dimension, and geometry) appears noteworthy. However, a significant revision is required to clarify the explanation of the proposed architecture.

---

### Official Review · Reviewer_BMSv · 2025-10-29

**Soundness:** 3
**Presentation:** 3
**Contribution:** 2
**Rating:** 4
**Confidence:** 4

**Summary:**

The paper proposes a coordinate free and a dimension free way to learn PDEs: instead of learning in a fixed grid/coordinate system, it learns the dynamics from an intrinsic feature library built only from the metric, gradients, Laplacians, and gradient inner products, that are well defined on any Riemannian manifold and in any spatial dimension. Models trained in simple 1D settings transfer to 2D/3D, curved surfaces, and different boundary conditions. The authors build the invariant feature set $B$, learn F(B) from data, and then integrate with a coordinate free integrator, or for implicit PDEs learn a low  dimensional manifold in $B$ space and enforce it via a PINN when solving on a new domain.
The authors show that models trained on 1D data accurately reproduce dynamics in: (i) other 1D coordinate systems, (ii) 2D/3D Euclidean domains (spiral/scroll waves for FitzHugh–Nagumo/Barkley), (iii) curved manifolds (sphere, in geographic/stereographic coords), and (iv) non-trivial topology (annulus), with quantitative error reports.

**Strengths:**

(i) Principled invariance: the paper builds a coordinate and dimension free feature library, so the learned law depends only on intrinsic geometric operators, and not on the chosen coordinates. A coordinate free integrator lets a model trained in simple 1D settings transfer directly to new coordinates, higher dimensions, and curved manifolds by recomputing the same invariants with the target metric, (ii) Cross domain demos: the authors show successful transfer 1D to 2D/3D, across coordinate systems, and onto curved/topologically nontrivial domains, providing concrete evidence that the invariants generalize, (iii) The work covers both evolution and implicit PDEs, hence broadening the applicability.

**Weaknesses:**

(i) Restricted operator class (up to second-order spatial derivatives): the invariant library B is built only from fields, Laplacians, and gradient inner products, (ii) The learned manifold in B space does not encode which boundary/initial data make the problem well posed, which must be supplied separately when moving to new domains, (iii) Limited stability analysis: long horizon stability or higher order integrators are not explored, (iv) The 2D Navier–Stokes example assumes that the pressure field is known and the continuity equation is given, the rollout is short (10 Euler steps), so the full difficulty of coupled scalar vector systems and pressure recovery is not fully addressed.

**Questions:**

Suggestions: (i) Ablations that isolate what really matters:  remove each invariant family (\Delta\Psi, gradient inner products) in turn, vary feature normalization, train in 1D and compare to training directly in target 2D/3D, in order to attribute gains to the intrinsic library, rather than incidental choices, (ii) Robustness study: quantify sensitivity to metric/discretization errors by perturbing g, adding mesh noise, varying stencil order, and report transfer degradation, (iii) Augment the learned B manifold with BC tags, Dirichlet/Neumann/mixed, or a boundary operator penalty so the constraint carries boundary information, (iv) Upgrade numerics and analyze long horizon stability, (v) For incompressible NS, add a coordinate free pressure recovery by Poisson solve, enforce \nabla\!\cdot v=0 via projection each step, and report transfer when pressure is not given.

---

### Official Review · Reviewer_wQZM · 2025-11-01

**Soundness:** 3
**Presentation:** 3
**Contribution:** 2
**Rating:** 2
**Confidence:** 5

**Summary:**

This paper proposes a framework for coordinate- and dimension-agnostic learning of partial differential equations (PDEs). The main idea is to represent PDE operators using a set of coordinate-free invariant features derived from classical invariant theory and Riemannian geometry. A neural network is then trained to learn the time-evolution law from these invariant quantities, enabling the model to generalize across different coordinate systems, geometries, and spatial dimensions.

The approach aims to achieve “spatially liberated” PDE learning, where a model trained on data from one domain (for example, a one-dimensional Cartesian grid) can be applied to others (such as curved or higher-dimensional manifolds) without retraining. The method is demonstrated on several benchmark systems, including FitzHugh–Nagumo, Barkley, Patlak–Keller–Segel, Helmholtz, and Navier–Stokes equations.

The paper claims three main contributions:
(i) a coordinate- and dimension-free feature representation for PDEs,
(ii) a neural framework for learning invariant operators, and
(iii) empirical demonstrations of geometric and dimensional transfer of the learned dynamics.

**Strengths:**

Originality:
The paper introduces a geometric formulation of PDE learning that enforces coordinate and dimension invariance. While the mathematical tools are drawn from established invariant theory, the combination with neural PDE modeling provides a moderately original framing rather than a fundamentally new methodology.

Quality:
The experiments are competently executed on standard reaction–diffusion systems. The results qualitatively demonstrate that a model trained in one coordinate system can reproduce dynamics in another, though the evaluation remains limited to simple toy settings.

Clarity:
The manuscript is clearly written and well-organized. The figures are helpful, and the derivations are presented with sufficient detail for replication. However, the exposition occasionally overstates generality and interpretability relative to what is actually demonstrated.

Significance:
The proposed framework may stimulate further exploration of coordinate-invariant representations in scientific ML. Nonetheless, its current scope—restricted to low-order, autonomous PDEs with constant coefficients—significantly limits its impact for practical PDE modeling or neural operator research.

**Weaknesses:**

Limited generality of the framework.
Despite claiming coordinate- and dimension-agnostic PDE learning, the method is applicable only to PDEs involving first-order time derivatives and up to second-order spatial derivatives. This excludes many physically and scientifically important systems such as the Cahn–Hilliard, Kuramoto–Sivashinsky, Swift–Hohenberg, and wave equations. The paper briefly mentions that higher-order operators could be included by extending the invariant basis, but this is not implemented or analyzed. A practical next step would be to demonstrate how the feature basis can systematically scale to higher derivative orders or nonlocal operators.

Restriction to autonomous, constant-coefficient PDEs.
The method implicitly assumes that coefficients are spatially uniform and that the PDE operator does not depend explicitly on time. This limits applicability to homogeneous systems. Extending the invariant basis to include spatially varying fields (e.g., variable diffusivity or forcing terms) or explicit time dependence would make the framework more relevant to real-world physical systems.

Overstated interpretability.
The paper repeatedly claims “physical interpretability,” but the neural network mapping from invariant features to time derivatives remains a black box. Using physically meaningful inputs does not make the learned operator interpretable in a mechanistic sense. To move toward genuine interpretability, the authors could integrate symbolic regression (e.g., SINDy, DeepMoD) or sparsity-promoting methods to recover explicit PDE forms from the learned operator.

Weak empirical validation.
All experiments involve simple synthetic systems (FitzHugh–Nagumo, Barkley, Patlak–Keller–Segel, Helmholtz) trained on one-dimensional data and evaluated on idealized higher-dimensional domains. There is no quantitative comparison with strong baselines especially recent development in symbolic machine learning for PDE discovery that already demonstrate high efficiency. Including quantitative benchmarks (e.g., error metrics, stability over time integration) and ablation studies on the invariant basis would substantiate the claimed generalization.

Inflated claims of dimension and geometry generalization.
Although the method generalizes from 1D to 2D or 3D manifolds, these extensions are limited to cases with identical underlying dynamics and simple coordinate transformations. The framework does not demonstrate robustness to complex topologies, irregular meshes, or noisy real data. A more realistic validation on non-ideal geometries (e.g., curved or anisotropic manifolds) would strengthen the generalization claim.

**Questions:**

Extension beyond second-order derivatives.
The current invariant basis is limited to first-order time and second-order spatial derivatives. Could the authors clarify whether their invariant construction extends in a systematic way to higher-order PDEs (e.g., fourth-order Cahn–Hilliard, Kuramoto–Sivashinsky)? Are there computational or theoretical barriers that prevent this extension?

Handling variable coefficients and inhomogeneities.
The proposed framework appears to assume constant coefficients and homogeneous domains. How would the method incorporate spatially varying coefficients (e.g., position-dependent diffusivity) or explicit time dependence? Would introducing these as additional scalar fields preserve coordinate invariance?

Scope of generalization.
The paper claims dimension- and geometry-agnostic learning, but the examples mostly involve smooth coordinate transformations (1D → 2D Cartesian or spherical). Can the authors clarify whether the framework can handle more complex manifolds (e.g., irregular meshes, topologically nontrivial domains, anisotropic metrics)?

Comparison with neural operator baselines.
Have the authors compared the proposed method with established symbolic machine learning approaches like deep symbolic regression, PySR, Finite Expression Method?

Interpretability claim.
The paper emphasizes physical interpretability, but the learned mapping F(B) is still a neural network with no transparent internal structure. Can the authors provide evidence that the model yields interpretable or analyzable representations (e.g., through symbolic regression, sparsity, or feature saliency)? If not, would they consider revising the claim to “physically consistent” rather than “interpretable”?

Quantitative evaluation and error metrics.
The experiments are primarily qualitative. Could the authors report quantitative metrics (e.g., relative L2 or energy norm errors) for predictions across domains and compare these to direct numerical simulations or neural operator baselines?

Effect of noise and data sparsity.
Real-world data are often sparse or noisy. The paper briefly mentions smoothing but does not explore robustness. How does the invariant representation behave under noise, interpolation errors, or partial observability of fields?

Computational cost and scalability.
The invariant construction and coordinate-free integration may add overhead compared to standard PDE solvers or neural operators. Can the authors discuss computational complexity and memory scaling when applied to high-resolution or multi-field systems?

Autonomous vs. non-autonomous systems.
The current formulation focuses on autonomous dynamics. Could the authors confirm whether the approach supports non-autonomous PDEs (e.g., external forcing or time-varying boundary conditions) and, if so, how time-dependent features would be integrated?

Clarification of the claimed “dimension-agnostic” property.
The notion of “dimension-agnostic” could be interpreted in multiple ways. Is the learned operator truly dimension-independent (i.e., reusable across dimensions without retraining), or is it empirically adjusted for each target dimension through retraining or fine-tuning? Clearer experimental evidence would help substantiate this claim.

---

### Author Response · Authors · 2025-11-29
**Revision Summary**

We thank all reviewers for their constructive feedback and for acknowledging the novelty of our approach. In response, we have revised the manuscript to clarify scope and assumptions, and we have added forward references to additional experiments already included in the appendices.

Specifically:
- Appendix K reports quantitative evaluations of transfer accuracy across dimensions and different geometries/topologies;
- Appendix L demonstrates robustness to realistic experimental noise;
- Appendix M shows boundary-condition transfer using a coordinate-free formulation;
- Appendix N provides an example of handling variable coefficients and symbolic extraction; and
- Appendix O illustrates the extension of our invariant feature library to higher-order PDEs, including a 4th-order example (Cahn–Hilliard).

These results collectively confirm that our framework generalizes $\textbf{without retraining}$, only by recomputing invariant features, remains robust under noise, adapts to different boundary conditions, and can be extended to PDEs with variable coefficients and higher-order terms.
We also revised the abstract, introduction, and discussion to downscope interpretability claims to “physical consistency,” clarify that our current focus is second-order PDEs under local symmetry assumptions, and include a limitations paragraph outlining nonlocal PDEs and long-horizon stability as future work.
New text appears in the revised manuscript in blue.

---

### Meta-Review · Area_Chair_uoGe · 2026-01-08

**Summary:**

The paper introduces a coordinate-free and a dimension-free way to learn PDEs.
It builds by learning the dynamics from an intrinsic feature library built only from the metric,
gradients, Laplacians, and gradient inner products.
By doing so, the learned model trained in simple 1d settings transfer to higher dimensional, curved surfaces, and different
boundary conditions.

Reviewers have raised some concerns about the work mostly related to the scope of the paper and the empirical validation.
Authors have not provided rebuttals but updated the paper. However, it seems to me that
despite extra experiments on higher-order PDEs and some quantitative experiments, the paper needs some major
revision improving on the several points raised by the reviewers (I would suggest a major rewriting
that includes in the main paper some key points present in the appendices).

**Reviewer Concerns:**

outstanding:
- scope of the work
- clarity of the story

**Reviewer Scores:**

I am not able to answer this.

---

### Decision · Program_Chairs · 2026-01-26

Reject